

# Evaluating the performance of coupled snow-soil models in SURFEXv8 to simulate the permafrost thermal regime at a high Arctic site

Mathieu Barrere[1,2,3,4,5], Florent Domine[1,2,3,6], Bertrand Decharme[7], Samuel Morin[4], Vincent Vionnet[4], Matthieu Lafaysse[4]

[1] Centre d'Études Nordiques, Université Laval, Québec City, QC, Canada
[2] Department of Geography, Université Laval, Québec City, QC, Canada
[3] Takuvik Joint International Laboratory, Université Laval (Canada) and CNRS-INSU (France), Québec City, QC, Canada
[4] Météo-France – CNRS, CNRM UMR 3589, CEN, Grenoble, France
[5] Univ. Grenoble Alpes - CNRS - IRD, IGE, Grenoble, France
[6] Department of Chemistry, Université Laval, Québec City, QC, Canada
[7] Météo-France – CNRS, CNRM UMR 3589, Toulouse, France

*Correspondence to*: Florent Domine (florent.domine@gmail.com)

**Abstract.** Global warming projections still suffer from a limited representation of the permafrost-carbon feedback. Predicting the response of the permafrost temperature to climate changes requires accurate simulations of the Arctic snow and soil properties. This study assesses the capacity of the coupled models ISBA-Crocus and ISBA-ES to simulate snow and soil properties at Bylot Island, a high Arctic site. Field measurements complemented with ERA-interim reanalysis were used to drive the models and to evaluate simulation outputs. Snow height, density, temperature, thermal conductivity and thermal resistance are examined to determine the critical variables involved in the soil thermal regime. Simulated soil properties are compared with measurements of thermal conductivity, temperature and water content. The simulated snow density profiles are erroneous, because Crocus and ES do not represent the upward water vapour fluxes generated by the strong temperature gradients within the snowpack. The resulting vertical profiles of thermal conductivity are inverted compared to observations, with high simulated values at the bottom of the snowpack. Still, ISBA-Crocus manages to successfully simulate the soil temperature in winter. Results are satisfactory in summer, but the temperature of the top soil could be better reproduced by representing adequately surface organic layers, i.e. mosses and litter, and in particular their water retention capacity. Transition periods (soil freezing and thawing) are the least well reproduced because the high basal snow thermal conductivity induces too rapid heat transfers between the soil and the snow in simulations. Hence, global climate models should carefully consider Arctic snow thermal properties, and especially the thermal conductivity of the basal snow layer, to perform accurate predictions of the permafrost evolution under climate changes.



# 1 Introduction

The Arctic is warming at twice the average planetary rate (Sweet et al., 2015). As climatologic, hydrologic and biological systems are fully coupled (Hinzman et al., 2005), climate change affects each Arctic ecosystem (Post et al., 2009; Serreze et

al., 2000). Consequences are already observed, such as reduced sea ice extent and snow cover duration (Serreze et al., 2000), shifts in vegetation (Pearson et al., 2013), permafrost degradation (Smith et al., 2010), and faunal redistribution (Post and Forchhammer, 2008).

Permafrost degradation is of major concern because of its feedback on the global climate system. Indeed, large amounts of organic carbon are stored in perennially frozen soils because of the limited microbial activity (Jonasson et al., 2001). Recent

studies estimate that about 1300 Pg of soil organic carbon (SOC) are stored in permafrost (Hugelius et al., 2014), constituting one of the largest terrestrial carbon pool. As permafrost is warming and thawing occurs, SOC becomes available for microbial mineralization, resulting in the release of potentially very important amounts of greenhouse gases (GHG) to the atmosphere (Elberling et al., 2013; Schuur et al., 2015). This effect is considered as one of the strongest positive climate feedbacks and needs to be taken into account in global temperature predictions. However, the permafrost-carbon feedback has not been

included in the climate projections of the IPCC Fifth Assessment Report (Schaefer et al., 2014), so that current warming predictions may be significantly underestimated.

Considerable uncertainties remain in the SOC decomposition rate and associated GHG emission in this global warming context. One of the main reasons is that the rate and extent of permafrost thaw is not well quantified, preventing accurate estimates of the potential magnitude of the permafrost-carbon feedback. In particular, how the permafrost thermal regime will

respond to climate change is still poorly represented because of its high sensitivity to the properties of the surface. The snow cover and the vegetation type are the main local factors influencing the permafrost thermal regime (Sturm et al., 2001a; Zhang. 2005), by modifying surface energy exchanges. A snow cover acts as a thermal insulator by limiting soil winter cooling, but its insulating properties are highly variable and insufficiently detailed in global climate models (GCMs). These characteristics, and how they will change with climate, therefore need to be considered to successfully simulate the current and future

permafrost thermal regime.

We attempt to contribute to these aspects by investigating the capacity of soil multi-layer version of the land surface model ISBA (Interactions Soil Biosphere Atmosphere; Decharme et al., 2013) coupled with the detailed snowpack schemes Crocus (Vionnet et al., 2012) or to the simpler scheme ES (Boone and Etchevers, 2001; Decharme et al., 2016) to simulate snow and soil properties at a high Arctic location. Crocus is currently the most detailed snowpack scheme coupled to ISBA, and

corresponds to the highest existing level of complexity for snowpack models. However, its relatively large computation time limits its use in global climate models. ES (Explicit Snow) is a multilayer snowpack scheme of intermediate complexity. Based on Crocus parameterizations, ES will be used for the upcoming CMIP6 CNRM-CM simulations (Eyring et al., 2016). Similar

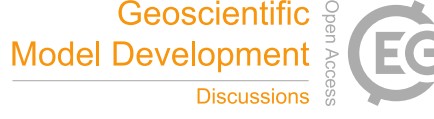

studies were performed over Siberian regions, using ISBA-Crocus (Brun et al., 2013) and ISBA-ES (Decharme et al, 2016). Models were evaluated based on snow height, snow water equivalent and soil temperature, but processes related to the simulated snowpacks have not been studied in detail yet.

Here, simulations are tested using field data obtained at one specific site on Bylot Island (73° N, 80° W), located north of
Baffin Island in the Canadian Arctic archipelago., The mean annual air temperature for the period 1998-2013 is -14.3°C and permafrost thickness has been estimated to be over 400 m (Fortier and Allard, 2004). Site description and instrumentation are given in the first section, after which the main features of the models are detailed. Simulation results are shown and compared to field observations. A sensitivity analysis is performed to determine processes critical to simulate Arctic snow and soil, which are discussed in the last section of the paper.

**2 Methods**

**2.1 Site description**

The study area is in the Qarlikturvik valley of the south-west plain of Bylot Island, around 73°10' N, 80°00' W (Fig. 1). The glacial retreat around 6000 years ago left fine grained wind-deposits and organic sediments to form the soil (Allard, 1996). The valley bottom consists in wetlands with typical permafrost landforms, including tundra polygons, thaw lakes and ponds.
Vegetation is mainly comprised of sedges, graminoids and mosses (Cadieux and Gauthier, 2008). Because uplands are better drained, mesic habitat dominates there with forbs, graminoids and erect vegetation. Erect vegetation is rather rare, covering less than 10% of the valley. It consists of willow shrubs (*Salix arctica* and *Salix richardsonii*), the higher shrubs being *Salix richardsonii* which commonly reaches 30 cm in height.

Our actual study site is located in wetlands of the valley floor, at 73°09'01.4" N, 80°00'16.6" W, in the middle of a low-center
tundra polygon. Vegetation consists of a typical herb tundra environment, covered by sedges, graminoids (*Poa* sp.), mosses (*Aulacumnium turgidum*, 2-3 cm thick) and small prostrate arctic willows (*Salix herbacea*). Albedo measurements were performed in July 2015, using a SVC HR-1024 spectroradiometer. Averaged over the 346-2513 nm spectral range, the surface albedo of our very site was 0.17 (M. Belke-Brea, personal communication, 2015). The soil granulometry was analysed using a laser scattering particle size analyzer, which resulted in a combination of fine grain deposits comprised of 64% silt and 36%
fine sand (Domine et al., 2016a), based on USDA classification. An organic litter layer (3.5 to 6 cm thick) was present at the interface between the surface vegetation and the ground. Because of the difficulty to determine the boundaries between the living moss, the litter and the underlying mineral soil, we estimated a total thickness of about 10 cm for both moss and litter. Field observations and simulations presented in this paper focus on this very spot.



## 2.2 Site instrumentation and data

As we investigate the coupled evolution of the atmosphere, snow and soil, various instruments have been installed to monitor meteorological conditions, along with snowpack and soil properties. The set of instruments was installed on the same tundra polygon, for the comprehensive monitoring of the spot.

Several automatic weather station (AWS) are operating on Bylot Island (CEN, 2013). In particular, a 10-m tower was installed in summer 2004 close to our study site, at 73°09'07.9" N, 79°59'19.0" W, to monitor air temperature and relative humidity, wind speed and direction and snow height. During summer 2013, an AWS (hereafter referred to as BylSta) was installed on the polygon of our study site (Domine et al., 2016a). It measures air temperature and relative humidity with a ventilated HC2S3 sensor from Rotronic, wind speed with a cup anemometer, surface temperature from an infrared IR120 sensor, and upwelling

and downwelling shortwave and longwave radiation with a CNR4 radiometer associated to a CNF4 heating/ventilating unit from Kipp & Zonen. These atmospheric variables are recorded hourly 2.3 m above the ground. An issue with the sensor caused erroneous air temperature measurements in the first year. This was fixed in summer 2014, providing a fairly complete meteorological dataset since then.

Snow height was automatically monitored with a SR50A acoustic gauge installed on the AWS. A few meters further, three

TP08 heated needle probes (NPs) from Hukseflux were placed at 7, 17 and 27 cm above the ground in summer 2013 to measure the snow temperature and its thermal conductivity $k_{snow}$. They were lowered to 2, 12 and 22 cm in July 2014 to better match the snowpack structure observed in May 2014. Operating methods and data analysis pertaining to the NPs are detailed in Domine et al. (2015), Domine et al. (2016a) and Morin et al. (2010). Applied to snow, the NP method is suspected of presenting a low systematic error of 20% on average, related to the granular structure of the medium (Calonne et al., 2011; Domine et al.,

2015; Riche and Schneebeli, 2013). The anisotropy of the snow structure is another possible source of error, because horizontally inserted NPs measure a mixture of vertical and horizontal thermal conductivities. As heat exchanges between the ground and the atmosphere occur in the vertical dimension through the snow, the anisotropy of the snow thermal conductivity can produce errors up to 20% in measurements, resulting in a maximum total error of 29% (Domine et al., 2015). Compared to the large range of $k_{snow}$ values (0.025-0.7 W m$^{-1}$ K$^{-1}$), and given that this method is the only suitable solution for remote field

work, errors related to the use of NP are acceptable. Furthermore systematic errors due to NPs may be subsequently corrected (Domine et al., 2015).

Field campaigns took place in May 2014 and 2015 at the end of the snow season. The snow accumulation was highly variable, because of wind effects and of microtopography (Liston and Sturm, 2002). The SR50A automatic spot measurements are therefore not necessarily representative of the average snow conditions. To explore the spatial variability of snow properties,

we performed hundreds of snow heights measurements covering different areas with an avalanche probe. In addition, snowpits were dug at a dozen specific sites to describe the stratigraphy, and to measure vertical profiles of density with a 100 cm³ box cutter, and temperature and thermal conductivity with a TP02 heated needle. Methods are detailed in Domine et al., 2016b. As we are focusing on the station site, we will only use data from the 3 snowpits dug within 20 m of the station both years.





Soil temperature and volumetric water content (VWC) have been monitored since July 2013 with Decagon 5TM probes, installed at depths of 2, 5, 10 and 15 cm. They were not calibrated for our specific soil, so we used the manufacturer's calibration for mineral soils which may produce an error in water content of 3%. The temperature sensor accuracy is within ±1°C. The active layer was 17 cm-thick at the time of installation so the sensors could not be placed deeper. A TP08 heated needle probe was inserted at 10 cm depth, just below the litter layer, to automatically monitor the soil thermal conductivity $k_{soil}$. The method used is the same as for the $k_{snow}$ measurements, and data analysis is also detailed in Domine et al., 2016a. In addition, two field campaigns were conducted in July 2014 and 2015 during which we measured soil water content profiles using an EC5 sensor from Decagon, and temperature and thermal conductivity with a TP02 heated needle at several dozen sites (Domine et al., 2016a).

### 2.3 Simulations

Simulations were performed using SURFEX (Surface Externalisée), the surface modelling platform developed by Météo-France and partners (Masson et al., 2013). Used in stand-alone mode, it takes meteorological data as driving input. The snow cover and the underlying soil are coupled through a semi-implicit scheme (Decharme et al., 2016; Vionnet et al., 2012). Different configurations are tested and compared with field observations. The results should evidence critical processes affecting the permafrost thermal regime.

### 2.3.1 Meteorological driving data

To calculate the energy and mass budget of the surface, the model needs the following input data: air temperature and humidity, wind speed, incoming shortwave and longwave radiation, precipitation rate (solid and liquid) and atmospheric pressure. We use observed local meteorological data when available, and missing data are filled with ERA-interim reanalysis (Dee et al., 2011). Available from 1979 to present, for a 0.7° grid with a 3-hourly time resolution, ERA-interim (ERAi) provides a continuous meteorological dataset available globally.

Air temperature and wind speed observations are available since 2004 from the 10-m tower. After July 2014, we used the ventilated air temperature and humidity, and the wind speed measured at BylSta. Radiation measurements were not used because the radiometer shifted by about 5° from its horizontal position when the tripod that supported it sank with the active layer thawing, causing errors to the data. The atmospheric pressure and precipitation were not measured, so we used ERA-interim reanalysis data for missing variables and possible data gaps.

To make the ERAi data consistent with the original field data, they were corrected following the method of Vuichard and Papale, 2015. This method consists in calculating the linear regression between ERAi data and available field measurements. The regression coefficients (slope and intercept) are used to correct systematic biases in ERAi data, in order to better match the local meteorological conditions. Hence, we found a mean bias of -2.3°C in air temperature, -4.7% in air humidity and +0.7

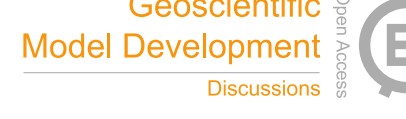

m s[-1] in wind speed given by ERAi. The correction led to reduce these respective biases by 20%, 3.3% and 10%. Differences in the correction performance mainly reflect the internal variability of the in-situ data.

The precipitation phase was recalculated using local temperatures with a threshold set at 1°C. During the ERAi reanalysis process, the precipitation fluxes are only predicted by the meteorological model with no assimilation of observations (Brun et al., 2013). Furthermore, our study site is surrounded by hills, altering the local aerology and therefore the precipitation amount. With no reliable precipitation gauge to evaluate the local variability, we consider the precipitation data as the highest uncertainty source of all the forcing data. Preliminary results indicated that an accurate snow height was critical to simulate correctly the soil thermal regime. Hence, we arbitrarily changed the ERAi precipitation data to match the observed snow height at snowpits dug in the immediate vicinity of BylSta, what was achieved after reducing by 30% the solid precipitation rate in winter 2013-2014. Neither liquid precipitations, nor snowfall were modified for the winter 2014-2015. ERAi radiation data are rather well correlated with the observations (r² = 0.8), with a mean bias of -8 W m[-2]. They were kept unchanged, as well as atmospheric pressure values which we assume suffer little from local variability.

### 2.3.2 Soil scheme

The land-surface parameterization is managed by the ISBA (Interactions between Soil Biosphere Atmosphere) scheme (Noilhan and Mahfouf, 1996; Noilhan and Planton, 1989). It describes the exchanges of energy and water between the atmosphere, the vegetation and the soil, by solving the 1-D Fourier's law for heat transfer and the mixed-form of Richards' equation for water mass transfer within the soil (Decharme et al., 2011). They are computed using 20 layers down to a depth of 12 meters. The main parameters are the soil texture and the vegetation type, from which other parameters can be derived, e.g. the soil porosity, the saturated matric potential and the saturated hydraulic conductivity. Following our measurements, the soil texture is set to 36% sand and 64% silt. Boreal grass covers the surface with a root depth of 20 cm, and the snow-free albedo is 0.17.

The soil freezing-thawing processes are critical in permafrost regions. They are handled using the drying-wetting analogy based on the Gibbs free-energy method (Boone et al., 2000). It allows the calculation of the temperature for phase changes as a function of the soil matric potential, which depends on porosity and water content. During phase changes, the liquid water content decreases (increases) correspondingly to the increase (decrease) in ice content, thereby conserving the total water content in each soil layer. ISBA calculates the hydrology of the entire soil column in order to accurately represent the permafrost characteristics. Therefore, the model needs long simulation periods to guarantee that the water and heat profiles are equilibrated over the 12 meters soil depth. Water infiltration is limited by the presence of ice-rich permafrost, and because of the topographic hollow of our study site water is at times poorly drained and can occasionally fill the center of the polygon. This process is reproduced by disabling lateral runoff when surface water is in excess.

The soil thermal conductivity is computed as a combination of water, ice and soil conductivities, volumetric water content and soil porosity (Peters-Lidard et al., 1998). Surface organic material, like moss covers or litter layers, are known to greatly affect the thermal and hydraulic properties of the soil (Hinzman et al., 1991). ISBA cannot handle a surface moss cover, but it has



the capacity to simulate a litter by conferring organic matter thermal properties to the uppermost soil layers. We ran a simulation including a 10 cm-thick litter which reduces the thermal conductivity of the first 10 cm below the surface. Based on our measurements, the litter appears to have a high insulating capacity in summer while it becomes negligible in winter. To better reproduce the observations, the litter effect is therefore disabled as soon as 2 cm of snow covers the surface. The soil

hydrology is not affected in the model, while in reality organic layers have a high hydraulic conductivity and a greater infiltration rate than the mineral soil (Hinzman et al., 1991). This will be evaluated with the results.

ISBA includes the dependency on soil organic carbon (SOC) content for hydraulic and thermal soil properties (Decharme et al., 2016). Accounting for the effect of SOC could significantly help improve the simulation of the soil thermal regime, especially in Arctic areas where soils store large amounts of SOC (Hugelius et al., 2014). From the Harmonized World Soil

Database (HWSD, FAO 2012) with a 1 km resolution, SOC content for our study site is estimated at 5.05 kg m$^{-3}$ for the first soil horizon (0–30 cm), and 4.34 kg m$^{-3}$ from 30 cm to 1-meter depth. A few soil samples were collected in July 2013 on the southern plain of Bylot Island for a radio-carbon analysis (ADAPT 2014). From two humid and three mesic sites, the carbon concentration was 5.17 kg m$^{-3}$ averaged over the first 30 cm of the soils. Given the excellent agreement with ADAPT data, we have a good confidence in the HWSD estimations.

### 2.3.3 Snowpack model: Crocus

Crocus is a multilayer physical snow scheme designed to simulate the evolution of the snow cover as a function of energy and mass transfer between the snowpack, the atmosphere and the ground (Brun et al., 1989; Vionnet et al., 2012). Numerical snow layers are handled dynamically by the model, in order to keep their total number below a given number (typically 50), while respecting as much as possible the internal structure of the snowpack and in particular strong contrasts between the properties

of distinct physical layers, and the fact that the discretized snowpack must comply with numerical constraints in particular the thickness of adjacent layers and the finer mesh size near the snowpack boundary. Numerical snow layers are characterized by their thickness, density, temperature, liquid water content, and four variables representing microstructural properties, i.e. dendricity, sphericity, grain size and one historical variable relating to the history of a given snow layer with respect to its past exposure to liquid water and strong temperature gradient (Vionnet et al., 2012). The freshly fallen snow is usually considered

as dendritic, and evolves toward non-dendritic snow under the action of internal processes such as snow metamorphism, compaction, thermal diffusion and phase change.

These processes also affect the density of the snow layers, which is a key variable controlling other snow physical properties in Crocus. Density increases with compaction caused by the weight of upper layers, depending on the viscosity of the layer. The compaction rate is faster in the presence of liquid water, and is also depends on snow grain type. Angular grains such as

depth hoar compact less. Wind events can also compact the surface layers, and due to frequent wind in the Arctic large amounts of snow are transported, compacted and sublimated (Liston and Sturm 2002; Sturm et al., 2001ab). Blowing snow occurs in the simulation when the wind speed is above a threshold value which depends on surface snow properties: microstructure and density (Guyomarc'h and Merindol, 1998; Vionnet et al., 2012). On average, blowing snow is observed for wind speed greater



than 5 m s[-1] (Sturm et al., 2001b; Vionnet et al., 2013). From the 10-m tower in Bylot Island, wind speeds greater than 5 m s[-1] occurred during 7% of the 2013-2014 snow season, and during 6% of the 2014-2015 one. Crocus takes into account the compaction and the microstructural evolution of the surface snow caused by blowing snow events, depending on wind speed, snow density and its microstructural properties (Vionnet et al., 2012). In addition, Crocus calculates the snow mass lost by

sublimation during blowing snow events (Brun et al., 2013; Gordon et al., 2006). However, Crocus does not handle the snow redistribution since the model is one-dimensional (Brun et al., 2013). In the standard version of Crocus, wind-compacted snow layers can reach a maximum density of 350 kg m[-3]. As we observed snow densities up to 450 kg m[-3] in Bylot Island, and values up to 600 kg m[-3] can be found in the literature (e.g. Sturm et al., 1997; Zhang, 2005), we increased this value to 600 kg m[-3]. Because the mobility of snow layers depends on their density, increasing the density of the snow can limit its driftability.

The snow thermal conductivity $k_{snow}$, in W m[-1] K[-1], which is used to solve the thermal diffusion equation in the snowpack, is calculated from the density using the equation of Yen (1981):

$$k_{snow} = k_{ice} \left(\frac{\rho}{\rho_w}\right)^{1.88} \tag{1}$$

with $k_{ice}$ the thermal conductivity of ice (2.22 W m[-1] K[-1]), $\rho$ the density of snow and $\rho_w$ the density of liquid water (1000 kg m[-3]).

Snow albedo depends on snow microstructure, on the amount of light absorbing impurities and on the solar zenith angle (Warren, 1982). Crocus computes the snow albedo by considering microstructure to account for physical aspects and the age of the surface snow layer to account for chemical aspects, as exposed snow is subjected to impurity deposition. The incoming radiation is then transmitted and absorbed within the snowpack, following the exponential decay of radiation with depth as a function of the grain size and snow density (Vionnet et al., 2012). Because incoming radiation is preferentially scattered

forward in the snow, light coming from a low sun angle will penetrate less deep in the snowpack and the resulting albedo will be higher. However, this last effect is not simulated in Crocus because it does not account for the solar zenith angle in the albedo calculation.

**2.3.4 Snowpack model: Explicit Snow**

Explicit Snow (ES) is a multilayer snowpack scheme of intermediate complexity (Boone and Etchevers, 2001; Decharme et

al., 2016). It is based on parameterizations used in Crocus, which are simplified to reduce the computation time and to facilitate its integration in global climate models. The main differences with Crocus are a constant number of layers, usually 12, blowing snow is not sublimated, and the snow microstructural properties are not simulated. Hence, parameterizations of snow albedo and compaction rate are function of the layers density only. However, as in Crocus, the evolution of snow density in each layer is due to snow compaction resulting from changes in snow viscosity and wind-induced densification of near-surface snow

layers (Decharme et al., 2016).



The $k_{snow}$ calculation method is different from Crocus. In ES $k_{snow}$ is computed from the density, based on the Yen's equation (Eq. 1), with an additional term to account for latent heat transfer through sublimation-condensation processes during metamorphism (Sun et al., 1999):

$$k_{snow} = k_{ice} \left( \frac{\rho}{\rho_w} \right)^{1.88} + \frac{P_0}{P_a} \times max \left( 0, k_1 - \frac{k_2}{T_s - k_3} \right) \quad (2)$$

where $P_a$ (Pa) is the air pressure, $P_0$ a reference pressure equal to 1000 hPa, $T_s$ (K) the temperature of the snow layer, and the coefficients $k_1$ = -0.06023 W m$^{-1}$ K$^{-1}$, $k_2$ = 2.5425 W m$^{-1}$, and $k_3$ = 289.99 K.

Even if ES is not based on the explicit representation of the snow microstructure, it remains one of the most sophisticated snowpack model that will be used for the upcoming 6th edition of the Coupled Model Intercomparison Project (CMIP6, Eyring et al., 2016). The results of our simulations with ISBA-ES are expected to reflect the capacity of the latest generation of GCMs

to simulate Arctic snow and soil properties.

### 2.3.5 Numerical experiments

A first simulation was run from August 1979 to August 2012, constituting a 33 years initialization. This operation allowed reaching equilibrium between the ground properties and the local climate conditions, with the *base* configuration. Then, using the equilibrated soil profile as the initial state, we performed several runs with different configurations from August 2012 to

June 2015:

- *base*: base simulation using SURFEX version 8.0 (rev. 4006), for a mineral soil.
- *litter*: addition of a 10 cm-thick surface litter.
- *SOC*: addition of organic carbon within the soil profile.
- *wind*: increase the maximum density for wind-induced snow compaction, from 350 to 600 kg m$^{-3}$.
- *ES*: use the ES snow scheme instead of Crocus.

Model configurations corresponding to each run are iteratively modified. ES uses the same configuration as the run *wind*.

### 2.4 Evaluation metrics

Field observations of snow and soil properties are available from August 2013 to June 2015, allowing the evaluation of model performance to simulate two winters and one entire summer.

Simulated snow properties are compared to measurements of snow height, density, thermal conductivity and temperature. As we assess the ability of the model to reproduce the soil thermal regime, we particularly focus on the snow thermal properties. Snow is thermally characterized by its thermal conductivity $k_{snow}$. We also rely on an alternate variable that characterizes the thermal properties of the whole snowpack rather than those of each layer. The thermal resistance of the snowpack $R_T$ (in m² K W$^{-1}$) depends on the thickness $h_i$ and the thermal conductivity of each layer:

$$R_T = \sum_i \frac{h_i}{k_{snow,i}} \quad (3)$$

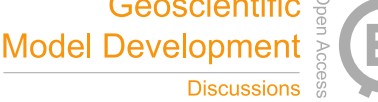



The simulated $R_T$ is computed from the respective density-$k_{snow}$ correlation used in the snowpack model (Eq. 1 and 2) and simulated layer thickness, while observed $R_T$ is directly based on measurements of $k_{snow}$ and layer thickness. To determine the respective contributions of the snowpack thickness and the thermal conductivity on $R_T$, it is also necessary to look at the resulting mean thermal conductivity of the snowpack calculated as:

$$\overline{k_{\text{snow}}} = \frac{h}{R_T} \qquad (4)$$

ISBA simulation results are compared to measurements of soil temperature, as well as the thermal conductivity and the water content which governs the soil thermal regime. Vertical profiles show the vertical soil stratification. Performances of each run are evaluated by comparing their deviations with time-series observations at different depths, using the squared correlation coefficient ($r^2$), bias (Eq. 5) and RMSE (Eq. 6) statistical errors.

$$bias = \frac{1}{n}\sum_i^n (\text{sim}_i - obs_i) \qquad (5)$$

$$RMSE = \sqrt{\frac{1}{n}\sum_i^n (\text{sim}_i - obs_i)^2} \qquad (6)$$

where *sim* and *obs* refer to simulated and observed values respectively.

## 3 Results

### 3.1 Simulations of snow properties and comparison with field data

### 3.1.1 Snow height

Figure 2 shows snow height automatically measured with the SR50A snow gauge, along with manual observations performed in May 2014 and 2015 and simulations results. SR50A data are missing between 14 May and 26 July 2014 because of an issue with the sensor. Random snow height measurements were performed within 200 m of our site on 14 and 16 May 2014, and on 12 May 2015. They evidence the large spatial variability in snow height, as already indicated in Table 2 of Domine et al., 2016b. Snow-free areas were frequently encountered on the polygons edges, whereas snow heights up to 60 cm were found in the center of the polygons. Thus, one-point snow height monitoring cannot be representative of the average snow accumulation. On 14 May 2014 the SR50A indicated 13 cm of snow, while we obtained a mean of 16.2 cm with a standard deviation of 13.7 cm from more than 300 measurements over the whole polygons area. We observed a noticeably lower snow accumulation under the station than a few meters away, which is confirmed by the 21 cm of snow averaged on the 3 snowpits dug in the vicinity of the station. On 12 May 2015, while the snow gauge indicated 35 cm, random measurements indicated 25.3 ±13.1 cm of snow. From the 3 snowpits made, we obtained a mean snow height of 33 cm. Based on our observations, we explain these differences by a low snow year in 2013-2014 and a high wind redistribution (Domine et al., 2016b). Since a central objective of our simulations is to reproduce the ground thermal regime as measured by our instruments, and because the large spatial variability in snow height highly affects the ground temperature at the meter scale (Gisnas et al., 2016), the relevant snow height at our specific site is most likely given by the snowpits mean.



After reducing the ERAi snowfall amount by 30% in winter 2013-2014, the resulting simulated snow height is in good agreement with the snowpits means obtained in May 2014 and 2015, respectively 23.0 and 30.4 cm from the run *wind* and 22.4 and 32.8 cm with ES. Winter variations are well reproduced, but the beginning of the snow season occurs too early in 2013: on 18 September in simulations while it was detected on 12 October from measurements. At this date the simulated snowpack was already 4 cm-thick, which partly explains the slight overestimation compared to the mean height of May 2014 snowpits. ES simulates lower snow heights than Crocus. It seems to be caused by a faster compaction rate in ES compared to Crocus. In May 2015, snowmelt seems to occur too early in simulations compared to automatic measurements. A detailed analysis of the meteorological data and simulation outputs reveals positive air temperature between 6 and 11 May 2015, triggering the partial melting of the upper snow layers in simulations. With the two strong wind events (peaks exceeding 10 m s$^{-1}$) which occurred on 5 and 12 May, these conditions caused the decrease of the simulated snow height along with the increase in the upper layers density. The air temperature cooled back down after this event, until 18 May when the final snowmelt started. Because we did not observe any signs of spring melt before 18 May during the field campaign, we found that this simulated early partial melting is caused by the lack of solar zenith angle consideration in the albedo calculation. Hence, for further comparisons with snow properties measured in May 2015, we will use simulations results of 6 May, just before the partial melting.

### 3.1.2 Snow density and thermal conductivity

Figure 3 shows examples of snow stratigraphies observed close to the monitoring site during the 2014 and 2015 field campaigns. They are typical of an Arctic snowpack (Domine et al., 2016b), comprised mainly of a basal depth hoar layer (5 to 10 cm thick) and a top wind slab. The particularity in 2015 was the indurated depth hoar layer at the bottom of the snowpack, resulting from the transformation of a melt-freeze layer into depth hoar under a very high temperature gradient (Domine et al., 2016a, b). Indurated depth hoar is harder and retains higher cohesion than typical depth hoar, but its development usually goes along with a decrease in density and thermal conductivity (Domine et al., 2012).

Associated measured vertical profiles of density are also shown. As expected from typical Arctic snowpack, the stratigraphies observed exhibit low density values for the bottom depth hoar layers (between 150 and 200 kg m$^{-3}$), and high values for the upper wind slabs (exceeding 400 kg m$^{-3}$). Crocus and ES simulate inverted profiles, with generally decreasing densities from the bottom to the top of the snowpack. This is particularly visible in 2015, with the highest density simulated at the bottom of the snowpack. In 2014 the density of the bottom layer (below 5 cm) is successfully reproduced by Crocus, but it is overestimated by ES. In 2015, the density of the indurated depth hoar basal layer is greatly overestimated by both Crocus and ES (310 kg m$^{-3}$ for the run *wind* and 374 kg m$^{-3}$ for the run *ES* compared to the mean observed of 181 kg m$^{-3}$). It confirms that the snow compaction rate in ES is inappropriate to simulate a bottom depth hoar layer of low density. However, it helps to reduce the error in upper layers densities. Wind slabs densities in upper layers are underestimated in simulations, even for the run *wind* which takes into account higher maximum densities. In 2014, the mean density for layers between 10 and 20 cm is





404 kg m$^{-3}$ for observations, 202 kg m$^{-3}$ for the run *wind* and 228 kg m$^{-3}$ for ES. In 2015, observations indicate a mean of 394 kg m$^{-3}$ while the run *wind* predicts 197 kg m$^{-3}$ and ES 301 kg m$^{-3}$ for layers between 10 and 20 cm.

Consequently, mean snowpack densities are lower in simulations than in observations. Observed profiles from 3 snowpits in 2014 result in 286 ±87 kg m$^{-3}$, and Crocus simulations give 203 ±2 kg m$^{-3}$. In 2015, we measured a mean density of 294 ±58

kg m$^{-3}$, while simulations with Crocus range between 179 and 194 kg m$^{-3}$. ES simulates higher mean densities compared to Crocus because the snowpack is overall more compacted. It results in mean densities of 261 and 264 kg m$^{-3}$ in May 2014 and 2015 respectively, in good agreement with measurements. However, the layering do not match the observations and the greatly overestimated basal density is compensated by the underestimation in upper layers.

Since both Crocus and ES calculate thermal conductivity mainly from density, the $k_{snow}$ vertical profiles are also inverted, as

already detailed in Domine et al. (2016a). For example in early May 2015 (Fig. 4), we measured $k_{snow}$ = 0.028 W m$^{-1}$ K$^{-1}$ 2 cm above the ground while Crocus (run *wind*) indicates 0.24 W m$^{-1}$ K$^{-1}$ and ES 0.35 W m$^{-1}$ K$^{-1}$. At 24 cm, values obtained from measurements, Crocus and ES are respectively 0.28, 0.07 and 0.14 W m$^{-1}$ K$^{-1}$. Figure 5 summarizes the $k_{snow}$ measurements performed during field campaigns, as a function of the corresponding measured densities and snow types. Simulated values obtained on 14 May 2014 and 6 May 2015 from runs *wind* and *ES* are also shown. It confirms that high density values (>400

kg m$^{-3}$), mostly observed in wind slab layers, are neither reproduced by Crocus nor ES. For lower densities, it is well visible that density-$k_{snow}$ correlations (Eq. 1 and 2) are not appropriate for our dataset. The simulated $k_{snow}$ values are almost always higher than measurements, and accounting for the Sun et al. (1999) additional term (run *ES*) amplifies the error. The regression curve from Sturm and al. (1997) fits our measurements better, because this parameterization is based on Arctic and subarctic snows instead of focusing on alpine conditions. However for a given density, the $k_{snow}$ value can vary by a factor of 4 to 5

(Domine et al., 2016a; Sturm et al., 1997) so that density-$k_{snow}$ correlations cannot be used to accurately determine the thermal conductivity of Arctic snow. In fact, as demonstrated by Calonne et al. (2014), snow thermal conductivity depends on both density and microstructure (see their Fig. 4). Our Fig. 5 strikingly corresponds to their theoretical Fig. 4 where our depth hoar values correspond to their lower bound and our rounded grain snows (including wind slabs) correspond to their upper bound. Our data therefore confirms theoretical considerations which clearly demonstrate that parameterizing thermal conductivity as

a function of density only simply should not be done and can only lead to gross errors.

### 3.1.3 Snow temperature

The snow density and $k_{snow}$ are the variables controlling the heat transfer through the snowpack, but they are significantly erroneous in simulations. To evaluate the consequences on the temperature profile, Figure 6 presents the evolution of temperatures measured by the NPs at 2, 12 and 22 cm, the surface temperature measured by the infrared IR120 sensor and the

corresponding simulated temperatures. The NPs perform measurements at 5:00 (local time) every other morning and record temperature only then. For this reason, simulated temperatures are shown every two days at about the same time. Because of the models' output resolution (6 hours), simulated temperatures are actually shown at 7:00 local time, or 12:00 UTC.



The high variability in snow heights makes it difficult to estimate the actual snow accumulation around the NPs. But it appears that temperatures in the middle of the snowpack are reproduced best. This is well visible at 22 cm, with the temperature being very accurately simulated after January 2015 while the NP is definitely buried in snow. Since the simulated thermal conductivity profile is inverted, we expect simulated snow surface temperatures to be colder than measurements while bottom

temperatures should be warmer. To a first approximation, under steady state the temperature gradient is inversely proportional to $k_{snow}$. Since simulated $k_{snow}$ values are low near the top, the temperature gradient is expected to be greater and therefore surface temperatures lower than measured, given that simulated and measured temperatures are similar near the middle of the snowpack. With similar reasoning, simulated basal snow temperatures are expected to be higher than measured. This is indeed what Fig. 6 shows. The Crocus simulated snow temperatures are about 5°C warmer at the bottom and 10°C colder at the

surface of the snowpack than measured in winter (run *wind*). ES snow temperatures are generally lower than Crocus because the simulated snowpack is more conductive. This is most pronounced at the bottom of the snow, where the $k_{snow}$ overestimation is considerable (Fig. 4), resulting in a bias of -5°C compared to measurements in February.

An extended warm spell started on 16 March 2015, with the air temperature reaching -2.2°C on 17 March associated with a significant snowfall (Fig. 2). This warmed up the entire snowpack, and from that date until the onset of simulated snowmelt,

snow temperatures are well reproduced by the run *wind*. The snowmelt starts on 18 May, and as already discussed the snowpack is warming faster in simulations.

### 3.1.4 Sensitivity analysis on the snow thermal resistance

Properties of the snowpits studied in May 2014 and 2015 are summarized in Fig. 7, and compared to simulation outputs. In 2015, we used simulations of 6 May before the early melting. Since we are investigating the link between the soil temperature

and snow thermal properties, Figure 7 shows the snow height, the mean snow thermal conductivity $\overline{k_{snow}}$ and the resulting thermal resistance $R_T$ as given by Eq. (3) and (4). Given that the NP method induces a systematic error which underestimates $k_{snow}$ by about 20% on average, we also increased the measured $k_{snow}$ values by 20% (*modified $k_{snow}$*).

Litter and SOC effects on the snow properties are low. They are essentially due to differences of less than 1 cm in snow height. Accounting for higher densities during wind compaction affects both the height and $k_{snow}$. Overall, the snow height is well

reproduced in simulations, results are within the measured standard deviation obtained from snowpits. $\overline{k_{snow}}$ is overestimated for both years of simulations, resulting in lower simulated thermal resistance than measured. Compared to measurements, the run *wind* gives a mean thermal conductivity 46% higher in 2014 and 73% higher in 2015, while the respective $R_T$ are 24% and 38% lower. The error in $R_T$ is thus essentially induced by the $k_{snow}$ simulation, and especially by the overestimation of the basal layer $k_{snow}$ value. Because ES simulates very high $k_{snow}$ values at the bottom of the snowpack (Fig. 4), the error in $R_T$ is amplified.

The larger error in $\overline{k_{snow}}$ simulated the second winter can be explained by considering snow stratigraphies (Fig. 3). In May 2014, we observed a regular depth hoar basal layer whose density was successfully reproduced by Crocus. Its simulated $k_{snow}$ is still overestimated to 0.11 W m⁻¹ K⁻¹, while we measured less than 0.03 W m⁻¹ K⁻¹. In May 2015, signs of early season melt-



freeze were observed in the basal depth hoar layer, which was indurated. Crocus and ES do reproduce the partial melting of the snow at the beginning of the season, but cannot simulate the following transformation into depth hoar under high temperature gradients (Domine et al., 2016a). As the water vapour flux through the snowpack is not represented, the models still consider a refrozen layer at the bottom of the snowpack with a very high $k_{snow}$ (8 to 12 times greater than measured, Fig. 4). Hence, the mean simulated $k_{snow}$ is more overestimated in May 2015 than in 2014.

With precipitations adjusted, Crocus and ES manage to reproduce the observed snow heights at our specific site, but the simulated snow layer densities are erroneous. The related vertical profiles of $k_{snow}$ are inverted in simulations, with consequences on the snow thermal resistance and the resulting heat transfers through the snowpack. Errors are amplified in ES, mainly because of very high simulated $k_{snow}$ values at the bottom of the snowpack.

## 3.2 Simulations of soil properties and comparison with field data

### 3.2.1 Soil thermal conductivity

The monitoring of the soil thermal conductivity ($k_{soil}$) at -10 cm shows a bimodal distribution between frozen and thawed state (Fig. 8). In summer, $k_{soil}$ is around 0.73 W m$^{-1}$ K$^{-1}$ while it suddenly increases with freezing to 1.95 W m$^{-1}$ K$^{-1}$ on average (Domine et al., 2016a). Note that the noise in the experimental data is due to the low power used to heat the NPs, so that heating is minimal when the ground is frozen because of the high conductivity of frozen soils. ISBA simulated $k_{soil}$ from the run *wind* is shown at depths of 10 and 20 cm. A bimodal distribution is also visible with sharp transitions between thawed and frozen state. The mean $k_{soil}$ value at -10 cm is 0.36 W m$^{-1}$ K$^{-1}$ in thawed conditions, and 2.04 W m$^{-1}$ K$^{-1}$ in frozen sate. The winter value is close to the measured mean, but the summer simulated $k_{soil}$ is lower than observations. Because the transition between litter and mineral soil occurs around 10 cm below the surface, measurements probably give a mixture of litter and mineral $k_{soil}$ value, while the simulated value at -10 cm is that of the litter only. Simulated $k_{soil}$ at 20 cm depth is 1.27 W m$^{-1}$ K$^{-1}$ in summer, and 1.89 W m$^{-1}$ K$^{-1}$ in winter. Averaging values between -10 and -20 cm results in a summer mean of 0.82 W m$^{-1}$ K$^{-1}$, and 1.97 W m$^{-1}$ K$^{-1}$ in winter, in very good agreement with our measurements. $k_{soil}$ is also greatly dependent on the water content, so we need to look at the vertical profiles of soil properties to assess the stratification.

Figure 9 shows vertical profiles of soil properties ($k_{soil}$, temperature and VWC) averaged from two measured profiles and simulated on 29 June 2014 to a depth of 20 cm. It illustrates the improvements caused by the litter addition in simulations. In the absence of litter, the simulated $k_{soil}$ profile from the run *base* stays constant at around 1.36 W m$^{-1}$ K$^{-1}$. The 10 cm-thick litter reduces $k_{soil}$ in the first 10 cm, which is more consistent with our observations. However, we measured an increase in $k_{soil}$ with depth in the first 10 cm, from 0.19 W m$^{-1}$ K$^{-1}$ at -3 cm to 0.71 W m$^{-1}$ K$^{-1}$ at -10 cm, while the simulated $k_{soil}$ is constant (0.35 W m$^{-1}$ K$^{-1}$) through the litter layer. The same pattern is visible on the VWC profiles, attesting the water content dependence of $k_{soil}$. The lowest VWC value was found at -3 cm with 20% moisture, immediately increasing to more than 45% at -6 and -10 cm, while the simulated water content stays constant at 40% (runs *wind* and *ES*). The VWC difference at -10 cm is not sufficient to explain the $k_{soil}$ underestimation in summer, meaning that measurements of $k_{soil}$ are probably affected by


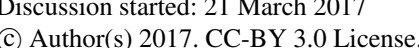


both the litter and the underlying mineral soil. The water content starts to decrease below -10 cm, which confirms the location of the lower limit of the litter. The presence of a litter in simulations finally improves the soil temperature profile between -10 and -20 cm deep, but the temperature is still too warm near the surface.

### 3.2.2 Soil temperature

Results of soil temperature simulations are shown in Fig. 10, along with measurements data from the Decagon sensors at depths of 5, 10 and 15 cm.

The run *base* simulates too warm temperatures in both summer and winter, and at the 3 depths shown. By reducing the heat exchanges between the atmosphere and the soil, litter and SOC additions greatly improve simulations in summer. Consequences are well visible at -15 cm, with a good simulation of the observed temperature (+0.8°C bias) during summer

2014 (June – July – August). For the same period, the temperature difference is +1.8°C at -5 cm, and +1°C at -10 cm. The same pattern is visible in Fig. 9, with a warm bias increasing toward the surface. The run *wind* improves the simulation of winter soil temperatures, resulting in the best simulation for the two years of measurements. Temperatures in winter months (December – January – February) are very well reproduced in 2014-2015, while in 2013-2014 there is a cold bias of 2.2°C in December, which becomes less than 1°C until snowmelt. ES produces soil temperatures up to 8°C colder in winter, because it

highly underestimates the snow thermal resistance (Fig. 7).

During the freezing and thawing periods, the models are not able to reproduce the observed soil temperature. The freezing of the active layer occurs in September and starts at the same time in simulations and observations, but it is too fast in simulations. The zero-curtain period, during which the soil temperature remains at 0°C while available water is freezing and latent heat is released, is lasting between 4 and 6 weeks in observations. In simulations, the soil totally freezes in a few days and cools too

early. It is because the thermal conductivity of the basal snow layer is too high in simulations, allowing the soil to cool rapidly. The thawing at the end of the winter is also faster in the simulations. During melting, the formation of a highly conductive refrozen snow layer is simulated at the top of the snowpack. For example in mid-May 2015, Crocus simulates $k_{snow}$ values up to 0.5 W m$^{-1}$ K$^{-1}$ at 25 cm, while we measured values ranging between 0.25 and 0.3 W m$^{-1}$ K$^{-1}$. Heat is thus easily transferred from the atmosphere to the soil in simulations. Ultimately, the treatment of snow albedo in models is such that melt-out is

accelerated, so that soil thawing is greatly accelerated in spring.

Table 1 summarizes the performance of each run to reproduce the measured soil temperature at 15 cm depth. Depths of 5 and 10 cm give similar results. $R^2$ are greater than 0.9, attesting the capacity of models to reproduce the observed variability. Statistics show a clear improvement as complexity is increased in the first 4 simulations. The presence of a litter noticeably improves the annual soil temperature simulation, while accounting for snow densities higher than 350 kg m$^{-3}$ (run *wind*)

drastically reduced the error in winter 2014-2015. The best results are obtained with the run *SOC* in 2013-2014 (+0.3°C for the whole year), and with the run *wind* in 2014-2015 (+0.5°C). Errors are very low for both years, of the order of the sensor accuracy.



### 3.2.3 Soil water content

Figure 11 shows the soil volumetric water content evolution at 5, 10 and 15 cm depths. In winter, the different runs have a low impact on the water content, which is very well reproduced with around 6% of water remaining liquid while temperature reaches -30°C. This effect is explained by surface tensions applied on water confined in small pores, lowering its freezing temperature (Penner, 1970). The 6% threshold observed here is consistent with values found in the literature for a mixture of silt and fine sand.

Litter and SOC additions improve the VWC simulation in summer, but it is still too low at -5 and -10 cm. Disabling the surface runoff helped to conserve high water contents in summer, but the moisture peak observed on 31 July 2014 at -5 cm is not simulated. On the ERAi precipitation data, a rain event did occur that day but it was less than 3 mm h$^{-1}$, not sufficient to increase the VWC by 20% as observed. Because of the high uncertainty on ERAi precipitation data, the actual precipitation rate could be underestimated. Figure 9 shows that the simulated VWC is constant in the first 10 cm of soil while the observed profile is more variable. Thus, it seems that the simulation of the water dynamics in the first cm of the soil is erroneous, and is better reproduced below -10 cm. There is also an offset on the phase changes timing between simulations and observations at depths of 10 and 15 cm. Compared to observations, the liquid water peak which follows the snowmelt occurs too early in simulations, and the freezing begins too soon. This is the consequence of inaccuracies in simulating the soil temperature during freezing-thawing periods. Phase changes are well reproduced for the 2013-2014 season at -5 cm.

Table 2 presents errors for each run in VWC simulation, at the depth of 15 cm. Results at depths of 5 and 10 cm are similar. R² are lower than those from temperature simulations, ranging from 0.39 for the whole year (run *base*) to more than 0.9 in winter. SOC addition greatly improves simulations. It results in the lowest errors for the two winters and the lowest bias for annual simulations. Errors in winter are lower than the sensor resolution (3%), while annual simulations give RMSE greater than 7%, reflecting the limit of the model to reproduce the summer moisture variability.

Despite the limitations of Crocus to reproduce the observed snow density profiles and the resulting thermal properties of the snow, the temperature of the soil is quite successfully reproduced in winter. The largest errors in temperature simulation are found during periods of phase changes, with too fast simulated soil freezing and thawing. Because ES simulates a very low snowpack thermal resistance, heat transfers are enhanced and the soil cools too much in winter. Soil properties are well reproduced at depths below 10 cm, but it seems there is a deficiency in the first cm below the surface.

## 4 Discussion

### 4.1 Snowpack simulations

In the Arctic, snowpacks are subjected to important upward water vapour fluxes generated by strong temperature gradients between the atmosphere and the ground. These fluxes lead to mass transfers from the lower (warmer) to the upper (colder) snow layers. Consequently, this process has a considerable contribution to snow metamorphism by decreasing densities at the



base of the snowpack and increasing densities of its upper parts (Domine et al., 2016a; Sturm and Benson, 1997). In snowpack models, the lack of representation of the vertical water vapour flux inevitably brings erroneous density profiles, which propagate to thermal conductivity values. The vertical $k_{snow}$ profiles are inverted in simulations, with high $k_{snow}$ at the bottom part of the snowpack and low $k_{snow}$ in the upper section, which affects the temperature gradient in the snowpack and the

boundary fluxes. Further, the absence of solar zenith angle consideration in the albedo calculation exaggerate the error in the surface heat flux in fall and spring, resulting in incorrect simulated melting episodes. The physically-based radiative transfer model TARTES is implemented in Crocus, making use of the solar zenith angle in the albedo calculation (Charrois et al., 2016; Libois et al., 2015). But the lack of field data on snow impurities (nature, deposition, light-absorbing spectroscopy) prevented us to use TARTES in Bylot Island simulations.

Despite their significant variability, averaged snow height obtained from snowpits in May 2014 and 2015 are well reproduced. To obtain that, the snowfall amount in winter 2013-2014 had to be reduced by 30%. Without this artificial modification, the models were not able to reproduce this low snow year. Given the importance of blowing snow events in the Arctic, accounting for snow redistribution by wind could also improve the simulated snow height (Gisnas et al., 2016; Libois et al., 2014). Even if relatively few strong wind events were recorded in Bylot Island in winter, Sturm et al. (2001b) showed that a single event

could transport important amounts of snow. Hence, snow compaction and sublimation caused by wind are also critical to simulate accurately snow height and density in the Arctic, which was already improved after increasing the maximum snow density to 600 kg m$^{-3}$. However, the models are not able to reproduce wind slab formation and the resulting increase in density as observed. But as already mentioned by Domine et al., 2016a: "The Crocus representation of the wind-packing process cannot be evaluated here, as the density increase also has contributions from water vapour deposition due to the upward flux

and their respective contributions cannot be observed separately."

### 4.2 Soil simulations

Granulometry is the main factor influencing soil properties, so it is important to analyze the soil composition of the studied site. Including SOC and especially litter greatly improved simulations of soil temperature and water content. Summer soil temperature is better reproduced at 15 cm depth than close to the surface. The addition of a litter improves simulation results,

but the model is still restrained by a limited representation of surface organic covers. A detailed analysis of measured and simulated vertical profiles of soil properties highlighted that the difference between the surface moss and the dead litter leads to a stratification in the first 2-3 cm of soil, with a lower thermal conductivity (i.e. a higher insulating capacity) for the moss. A moss surface also increases the water infiltration (Beringer et al., 2001), resulting in a dryer soil surface and a greater storage of moisture in lower layers. Because $k_{soil}$ and the freezing process are highly dependent on the water content, improving the

hydraulic properties of the soil scheme by considering a moss cover could help better reproduce the zero-curtain period and the soil thermal regime for layers close to the surface.

In light of the results of the snow-soil coupled evolution, it appears that errors in simulating the soil thermal regime manifest themselves mostly during freezing and thawing. The too rapid simulated freezing and thawing are due to the high thermal



conductivity of the basal snow layer. For thawing, this is enhanced because of the early simulated snowmelt, linked to inadequate albedo simulations, which have a critical impact in May. It is interesting that the most sophisticated model run (*wind*) is able to reproduce quite accurately the ground thermal regime at all depths in winter (Fig. 10), even though the thermal properties of the snow are not accurately simulated. Numerical models can be viewed as just the description of set of complex

processes where error compensation is optimized. Therefore, we suggest that the insufficient insulating properties of the simulated snowpack are compensated by the inverted thermal conductivity profile, because the simulated insulating top snow layer damps air temperature fluctuations and thus limits heat transfer during cold spells. Under periodic diurnal temperature fluctuations, cold waves cannot penetrate into the simulated snowpack as well as when a conducting layer is present, so that the overall heat loss would be reduced by the inverted stratification. Of course, this is just a hypothesis that needs to be tested

in future work, but some process exists, that compensates for the insufficiently insulating simulated snowpack, to explain the excellent soil temperature simulation most of the time.

### 4.3 Representativity of observations

The very large spatial variability of snow properties makes the comparison between simulations outputs and observations difficult. In particular the snow height appeared to vary a lot with the microtopography at the 50-cm scale, thus only

measurements performed in the immediate vicinity of the station can be used to assess the link between snow and soil properties measured at one specific site.

The lack of precipitation measurements is a major concern, because most of the snow and soil properties depend on the amount of precipitation. In particular, precipitation controls the snow height and the soil water content. The winter 2013-2014 was a particularly low snow year, with high wind redistribution. We thus arbitrarily changed the snowfall amount given by ERAi to

reproduce the mean snow accumulation observed at our site. It is well known that measuring precipitation in the Arctic is a challenge, especially in winter when snow falls in windy conditions and precipitation gauges are only catching between 20 to 70% of the actual amount of snow (Goodison et al., 1998; Liston and Sturm, 2002). But using a shielded precipitation gauge and correcting data using wind speed data should help reduce the large uncertainty that we have using ERAi precipitation data alone, as already recommended by Bokhorst et al., 2016.

**5 Conclusion**

Applying the coupled snowpack-soil models ISBA-Crocus and ISBA-ES to a high Arctic site reveals major deficiencies related to typical Arctic conditions. The main weakness lies in the simulation of snow physical properties, because the absence of a modelled upward water vapour flux prevents reproducing the observed density profiles. The resulting vertical profiles of $k_{snow}$ are inverted in simulations, producing erroneous heat transfers through the snowpack. This work also illustrates that

determining snow thermal conductivity from density only is totally inappropriate, especially (but not only) in the Arctic,



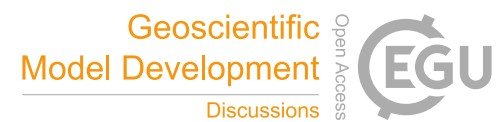

confirming the theoretical work of Calonne et al. (2014). Considering also a microstructural variable, or perhaps snow type, appears mandatory.

The soil temperature is the least well simulated during freezing and thawing periods. The main reason is the too conductive basal snow layer, which allows the soil to cool and warm rapidly. Still, ISBA-Crocus manages to reproduce the temperature

of the soil satisfactorily in summer and winter. Our results suggest that errors in the $k_{snow}$ stratification can compensate errors in simulated $k_{snow}$ values in winter. Hence, despite its apparent good results, Crocus is not better adapted than simpler models like ES to simulate Arctic snow thermal properties. The snow height has also a major influence on the winter soil temperature, but it is highly variable because of the wind-induced snow redistribution. Finally, a better representation of surface organic layers should improve simulations of the top soil properties, in particular the water content which controls the soil thermal

properties and the water phase changes. The water content also governs the amount of water vapour transferred from the soil to the snow.

There are therefore strong uncertainties in climate projections related to the permafrost-carbon feedback, because even the most sophisticated snow models cannot accurately simulate heat transfers through Arctic snowpacks. In particular the soil freezing and thawing processes need to be carefully simulated, because they determine most of the permafrost properties

(Hinzman et al., 1991). The dual challenge of GCMs is thus to improve the representation of snow Arctic processes, in particular the water vapour fluxes and the resulting density and thermal conductivity profiles, and to include sub-grid snow height heterogeneities (Liston, 2004) in order to accurately simulate the thermal regime of permafrost.

**6 Data and code availability**

Forcing data used for this research and evaluation data of figures 2, 6, 8, 10 and 11 are available for further use in the scientific

community at Nordicana D, doi:10.5885/45460CE-9B80A99D55F94D95. The strong differences between Arctic and alpine snowpack types encourage to incorporate such datasets in the upcoming model intercomparison exercises, in particular ESM-SnowMIP (http://www.geos.ed.ac.uk/~ressery/ESM-SnowMIP.html) which currently does not include sites featuring tundra snow types. The model is open source, it is available via the SURFEX platform which can be downloaded at http://www.umr-cnrm.fr/surfex/. The simulations presented in this paper were realized with SURFEX version 8.0, revision 4006.

*Author contribution.* FD and SM designed the research. MB and FD performed the field measurements and analyzed the data. MB realized the model simulations with advice from BD, SM, VV and ML. MB prepared the manuscript with contributions from all co-authors.

*Competing interests.* The authors declare that they have no conflict of interest.



*Acknowledgments.* This work was supported by the French Polar Institute (IPEV) through grant 1042 to FD, by NSERC through the discovery grant program and by the BNP Paribas Foundation. MB received scholarships from EnviroNord, and grants from Consulat Général de France à Québec and Région Rhône-Alpes. The Polar Continental Shelf Program (PCSP) efficiently provided logistical support for the research at Bylot Island. We are grateful to Denis Sarrazin for his precious

technical support on the CEN weather stations, and to Gilles Gauthier and Marie-Christine Cadieux for their decades-long efforts to build and maintain the research base of the Centre d'Etudes Nordiques at Bylot Island. Bylot Island is located within Sirmilik National Park, and we thank Parks Canada and the Pond Inlet community (Mittimatalik) for permission to work there. CNRM/CEN and IGE are part of LabEx OSUG@2020.

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

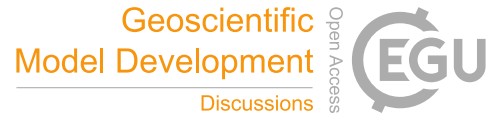

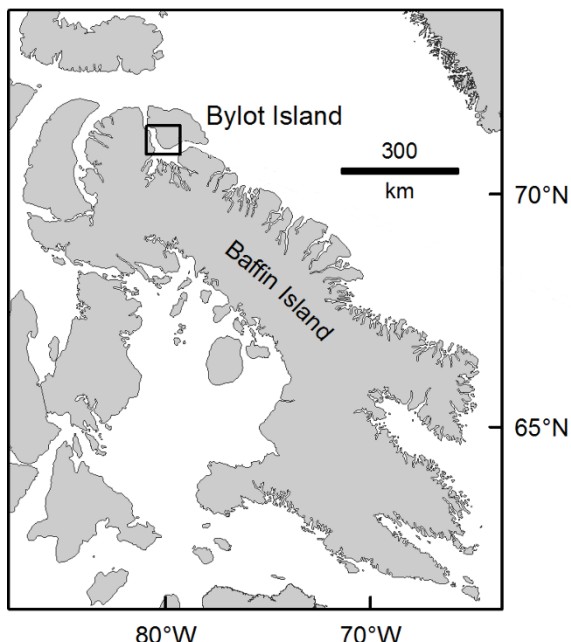

**Figure 1. Location of the study site in the south-west plain of Bylot Island, in the Canadian Arctic archipelago.**





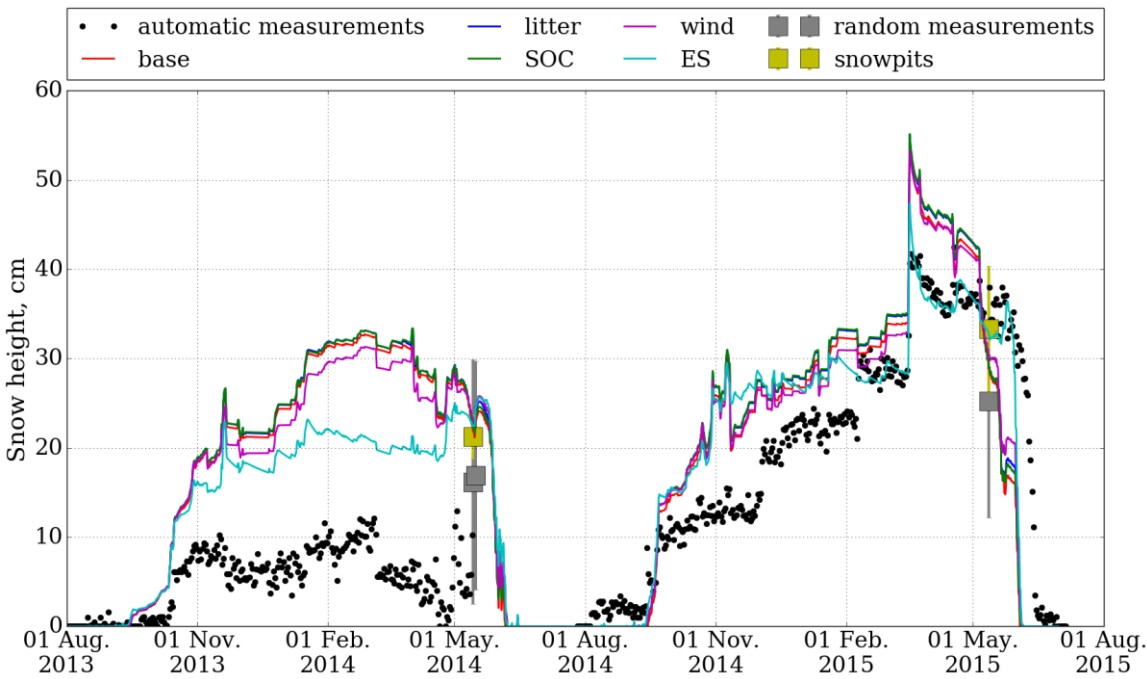

**Figure 2. Snow height evolution over the 2 years of observations. Results of field observations in May 2014 and 2015 are also shown.**



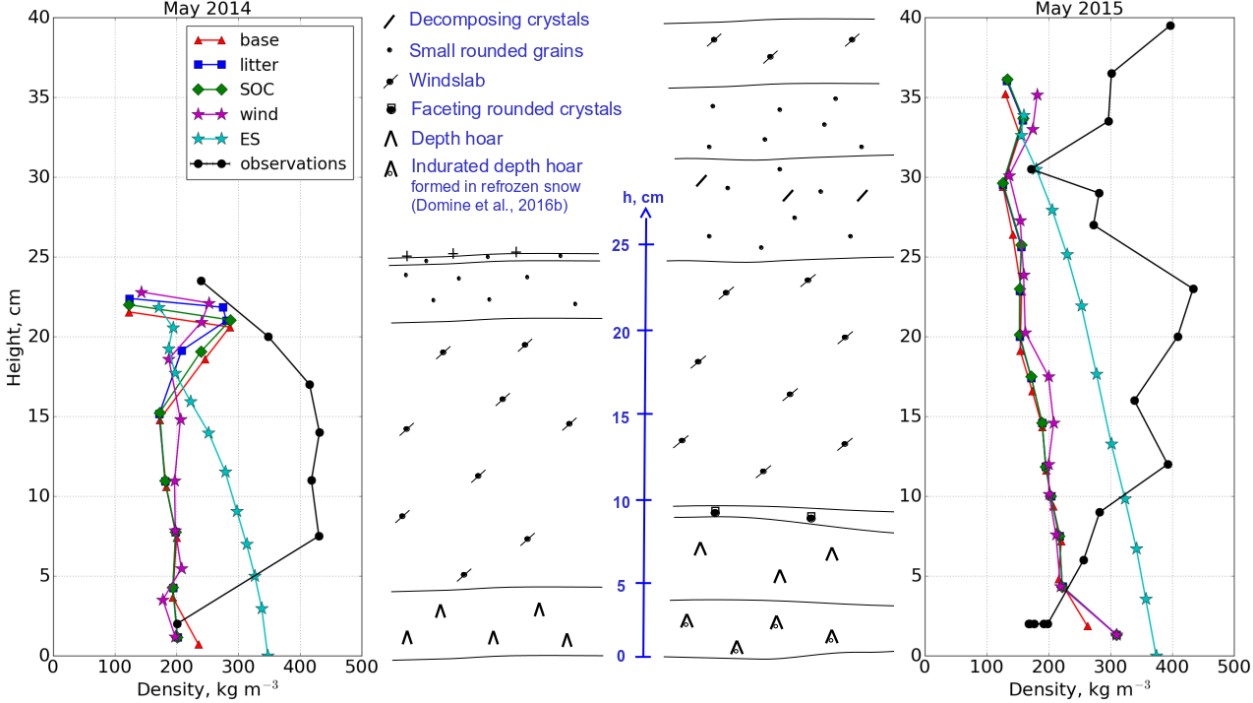

**Figure 3. Stratigraphies and vertical profiles of density measured on 14 May 2014 (left) and 12 May 2015 (right), and simulated densities on 14 May 2014 and 6 May 2015.**





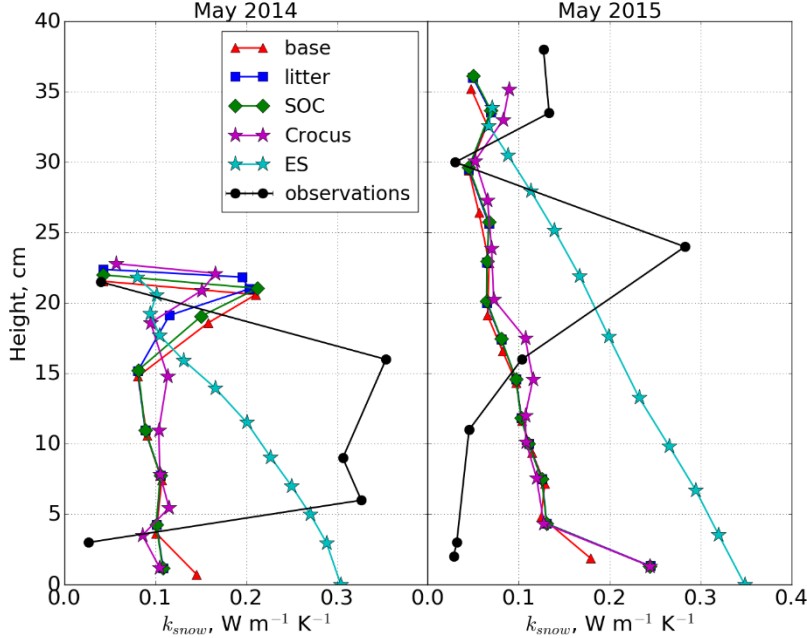

**Figure 4. Vertical profiles of snow thermal conductivities measured on 14 May 2014 (left) and 12 May 2015 (right), and simulated on 14 May 2014 and 6 May 2015.**





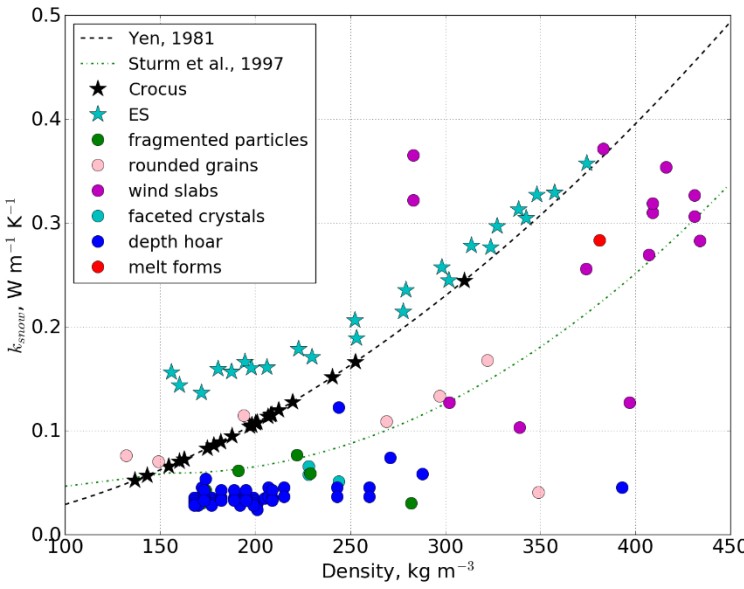

**Figure 5. Measured and simulated snow thermal conductivity values as a function of density. Regression curves from Yen (1981) and Sturm et al. (1997) are also shown, as well as observed snow types.**





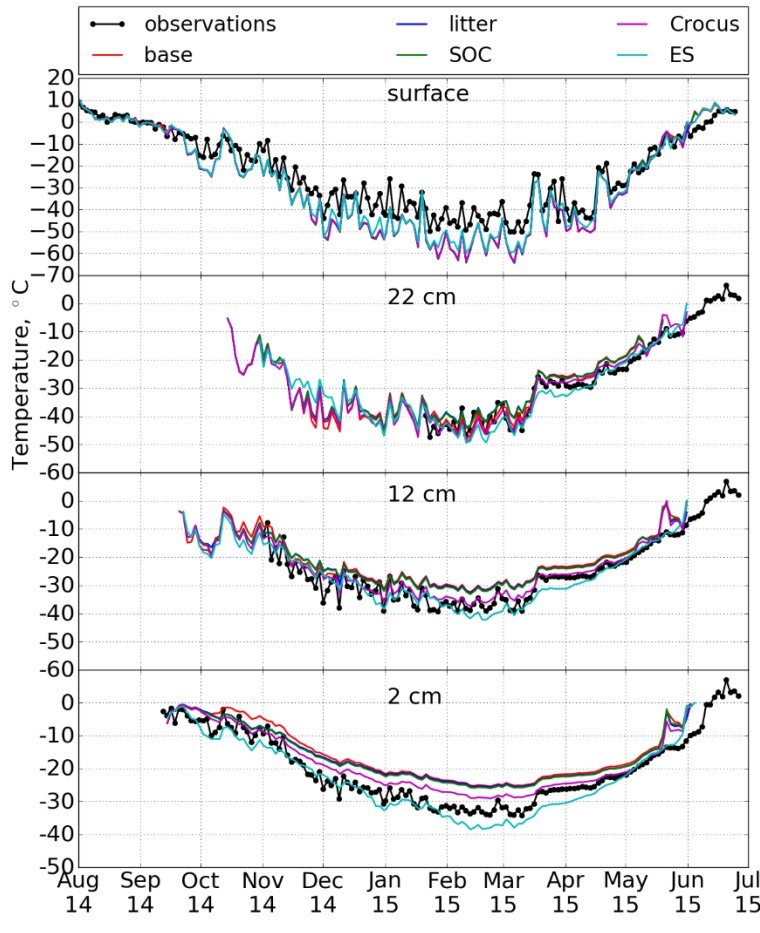

**Figure 6. Evolution of simulated and observed temperatures within the snowpack during winter 2014-2015. Observations at 2, 12 and 22 cm come from the NPs, and the surface temperature is measured by the IR120 sensor.**





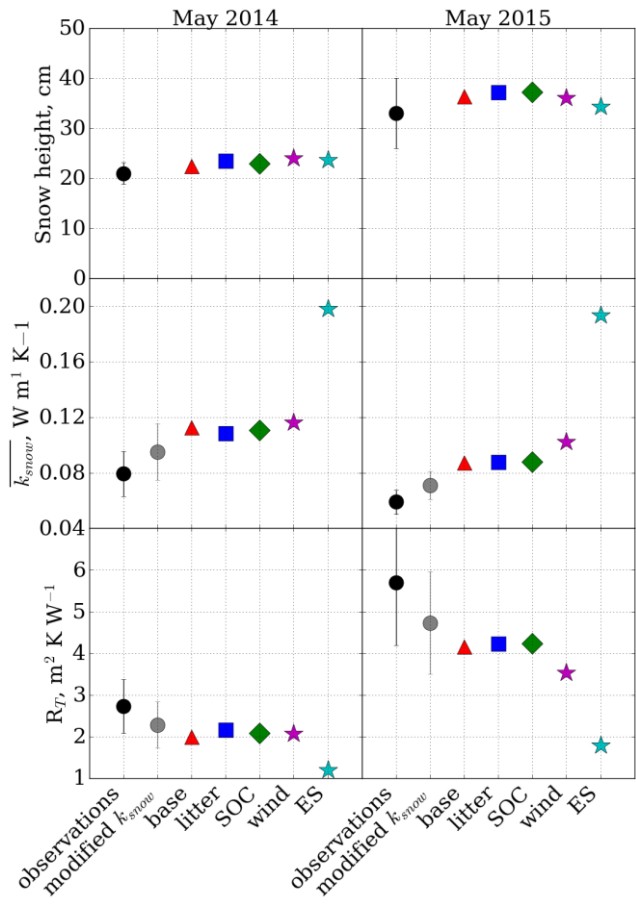

**Figure 7. Overview of snow heights (top), average thermal conductivities (middle) and thermal resistance (bottom) measured during May 2014 and 2015 field campaigns, and simulated values.**





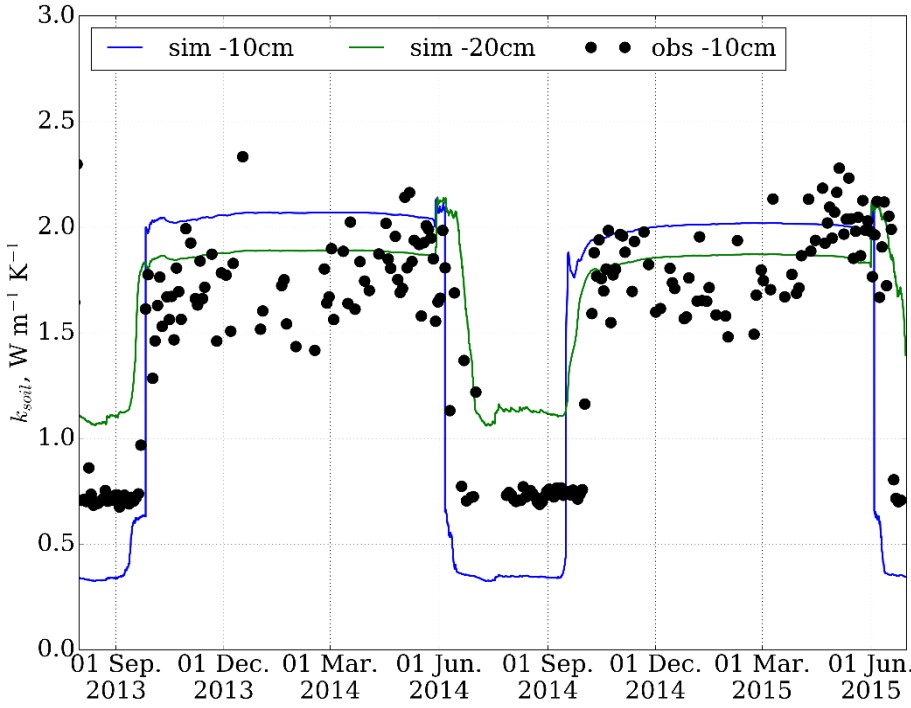

**Figure 8. Evolution of the soil thermal conductivity measured at 10 cm depth, and simulated (run *wind*) at depths of 10 and 20 cm.**



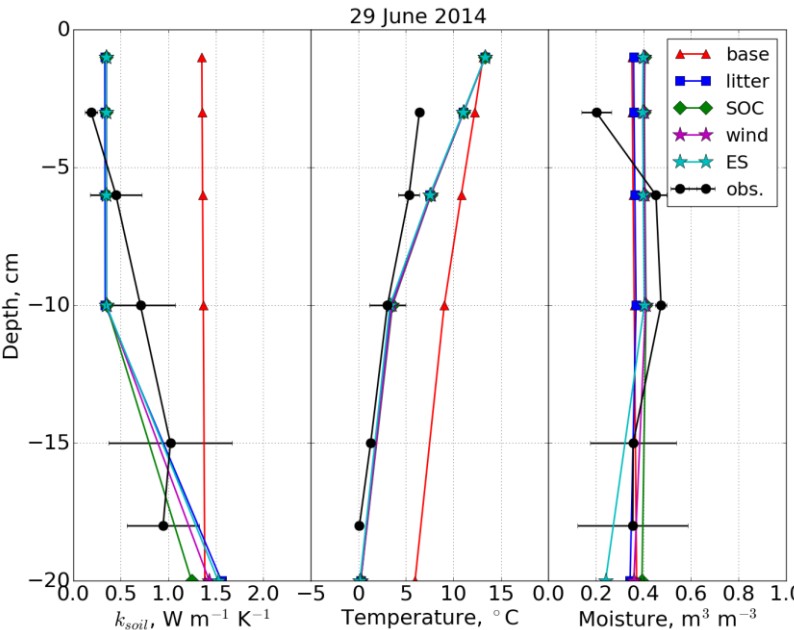

**Figure 9.** Vertical profiles of $k_{soil}$, soil temperature and volumetric water content measured on 29 June 2014 and simulated in the first 20 cm below the surface. Horizontal bars indicate the standard deviation of measurements.


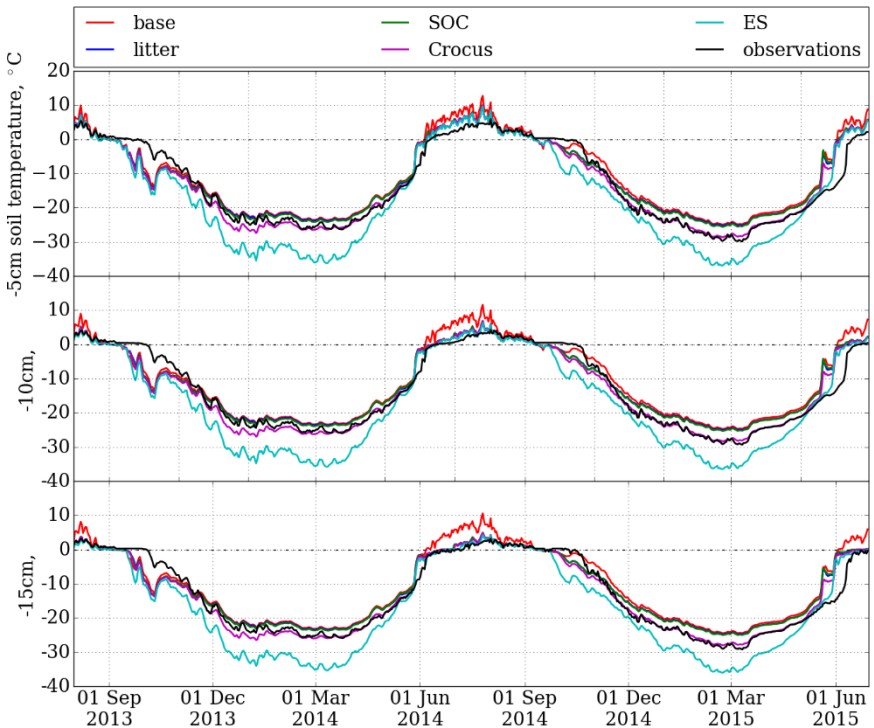

**Figure 10. Observed and simulated daily mean soil temperature at 5, 10 and 15 cm deep.**





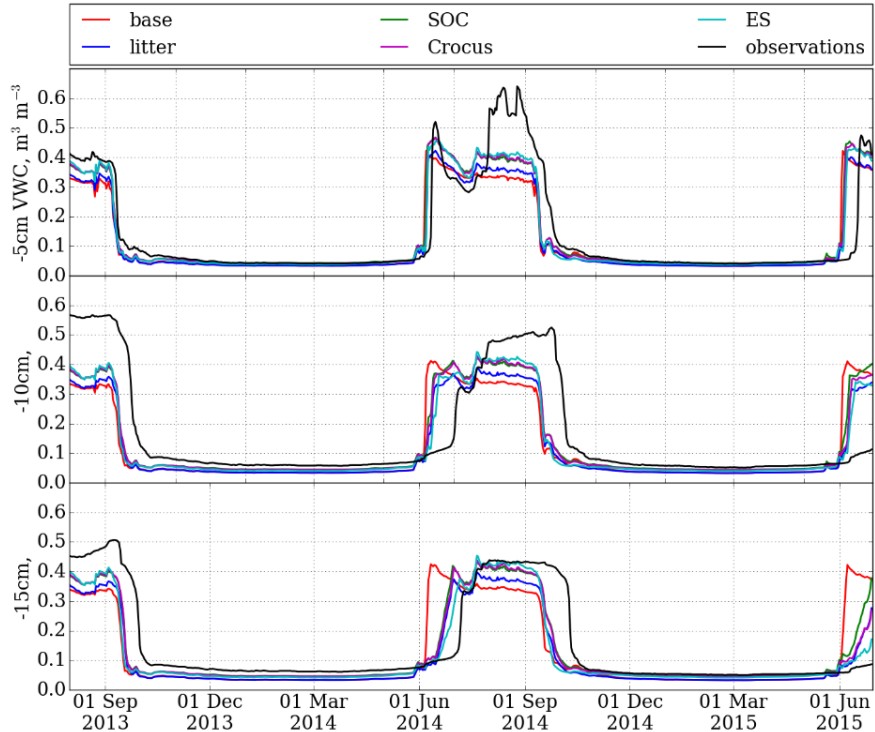

**Figure 11. Observed and simulated daily mean soil volumetric water content at depths of 5, 10 and 15 cm.**





**Table 1. Statistical indicators of the differences between observations and model results of the soil temperature (°C) at -15 cm on a 6-hour timestep. Bold values correspond to the best scores.**

| year | period | type | runs | | | | |
|------|--------|------|------|--------|-----|------|------|
| | | | base | litter | SOC | wind | ES |
| 2013-2014 | annual | bias | 1.48 | 0.37 | **0.32** | -0.92 | -4.42 |
| | | rmse | 3.41 | 2.62 | 2.61 | **2.54** | 5.91 |
| | | r² | 0.92 | 0.94 | 0.94 | **0.95** | **0.95** |
| | winter (DJF) | bias | 1.85 | 1.54 | **1.33** | -1.42 | -8.01 |
| | | rmse | 1.94 | 1.66 | **1.45** | 1.62 | 8.06 |
| | | r² | **0.94** | **0.94** | **0.94** | 0.88 | 0.93 |
| 2014-2015 | annual | bias | 3.31 | 2.08 | 2.04 | **0.50** | -2.96 |
| | | rmse | 4.34 | 3.49 | 3.52 | **2.63** | 4.86 |
| | | r² | 0.94 | 0.94 | 0.94 | **0.95** | 0.92 |
| | winter (DJF) | bias | 3.64 | 3.06 | 2.89 | **0.38** | -5.07 |
| | | rmse | 3.68 | 3.12 | 2.95 | **0.61** | 5.39 |
| | | r² | 0.97 | 0.97 | 0.97 | **0.98** | 0.96 |





**Table 2. Statistical indicators of the differences between observations and model results of the soil volumetric water content (%) at -15 cm on a 6-hour timestep. Bold values indicate the best scores.**

| year | period | type | runs | | | | |
|------|--------|------|------|--------|------|------|------|
| | | | base | litter | SOC | wind | ES |
| 2013-2014 | annual | bias | -3.18 | -4.18 | **-2.21** | -2.38 | -3.20 |
| | | rmse | 11.85 | 8.90 | 8.37 | 8.19 | **8.01** |
| | | r² | 0.48 | 0.73 | 0.72 | 0.74 | **0.77** |
| | winter (DJF) | bias | -2.97 | -2.99 | **-1.67** | -1.83 | -2.13 |
| | | rmse | 2.97 | 2.99 | **1.67** | 1.83 | 2.14 |
| | | r² | **0.93** | **0.93** | **0.93** | 0.85 | 0.90 |
| 2014-2015 | annual | bias | -2.31 | -3.66 | **-1.41** | -1.97 | -2.69 |
| | | rmse | 12.30 | 8.33 | 7.97 | **7.29** | 7.99 |
| | | r² | 0.39 | 0.75 | 0.72 | **0.78** | 0.74 |
| | winter (DJF) | bias | -1.81 | -1.85 | **-0.52** | -0.67 | -0.93 |
| | | rmse | 1.81 | 1.85 | **0.53** | 0.67 | 0.93 |
| | | r² | 0.89 | **0.90** | **0.90** | **0.90** | 0.89 |