# Peer review of "Evaluating the performance of coupled snow-soil models in SURFEXv8 to simulate the permafrost thermal regime at a high Arctic site"

_Geoscientific Model Development, 2017_

## Referee Comment (RC1) · Anonymous Referee #1 · 24 Apr 2017

[Figure]

This paper evaluates model skill in reproducing snow and soil characteristics at a permafrost site in Canada. At this site, a variety of in-situ data were retrieved: snow height, snow stratigraphy, temperature, and conductivity. The aim of the study is to find out how good a coupled soil-snow model is at reproducing the transient temperature signal at this site, during two consecutive seasons (2013-2015). Model runs have been performed with the Crocus snow model and the ISBA soil model. Sensitivity experiments are performed one snow property and several soil properties. In addition, the simpler ES snow model has been included, for comparison.

Printer-friendly version

[Figure]

I think this study identifies important shortcomings in these snow models, some of which will apply to other models as well. The article is well structured and contains material that complements existing studies with Crocus. Unfortunately, the paper is not particularly well written and tends to 'jump to conclusions' which undermines its academic quality. I recommend publication after the following issues have been addressed.

Major comments

The paper is too long in general. I believe it should fit on $\frac{3}{4}$ of the current number of pages. Leave out unnecessary sentences, e.g. only summarize site instrumentation, and refer to full discussion in Domine (2016a). Rewrite long sentences. More use of academic language which is shorter. Focus on main results. Suggest to combine Results and Discussion. Perhaps make two new sections out of those: (1) Snow model results and (2) Soil model results.

Grammar: the paper could have a better use of linking words to create flow and thus make it easier to read. There are grammatical errors that need to be fixed before publication. I listed some at Specific Comments but I did not aim for completeness.

Suggest to present the experiments in a table, instead of in the text. This would make them easier to refer to and would help to clarify the text, e.g. Sect. 2.3. Then, introduce the experiments earlier in the text, e.g. 'litter' on page 7 line 1, 'SOC' on page 7, line 8-14, 'wind' on page 8, line 8 . What confused me at first is that the additions are additions with respect to other experiments, not 'base'. It would be great if this could be made more clear.

One of the main results is Figure 3, that shows that snow density is not well reproduced by the models. It is hypothesized by the authors that this is due to a missing process: upward vapour fluxes. Yet they have no run with this process included so they cannot conclude that this process is fully responsible, only partially, or not at all. The abstract is therefore misleading (Page 1, line 20-22) and should be changed.

On page 8 line 8 the authors explain that compaction by drifting snow can now reach up to 600 kg/m3, compared to 350 kg/m3 before. The argumentation for this change is anecdotal ('we observed densities of 450 kg/m3'). Should the reader therefore regard this change as just a sensitivity test, rather than a real physical process that was misrepresented? Moreover, doing this you may be compensating for other biases / missing physics in the model, such as the missing upward vapour transport, and, my hypothesis, early melt and refreezing? This potential caveat is not discussed.

I guess the goal of simulation 'wind' is to simulate a hardened top wind slab. Rather than changing the upper limit to 600 kg/m3 in simulation 'wind', would it not be more effective to decrease the characteristic time scale in the drifting snow compaction (parameter Tau in Vionnet, 2012)? Looking at Figure 3, I see none of the model results exceed 300 kg/m3 at the top, so I wonder if 600 kg/m3 is ever reached at all.

In Figure 3, density in May 2014, the 'wind' experiment simulates lower density at the top than the others. This is counter-intuitive, as you would expect always higher densities in this simulation. Could you provide a possible explanation to why this is?

The analysis of snow temperature completely omits the effect of latent heat by rain and meltwater refreezing. What do the authors think is the importance of refreezing on temperature and how do the models simulate this?

Another key result is Figure 5, that shows that a simple density relationship for thermal conductivity is not sufficient to reproduce most observations. That said, it does not deserve the qualification 'totally inappropriate' (P18, L30).

P17, L5-6: the effect of missing effect of solar zenith angle is stated like a fact. Yet you have no results or reference to support this. Make clear that this is a hypothesis, not a given fact.

The two previous comment exemplify a general critique that I have on this article: the wording is not precise enough. In the article, there are sentences without such modi-

fiers that read like facts, but are in reality claims or beliefs of the authors. This must be addressed in the final version.

The authors do not mention whether the model changes they did (litter/SOC/wind) have officially been incorporated into SURFEX.

Specific comments

P1, L19: soil and snow thermal regime. Simulated soil and snow properties.

P1, L20: compared with → compared to, add comma after 'temperature'

P1, L21: suggest to change 'erroneous' to 'unrealistic'

P1, L29: climate change.

P2, L31: ES is introduced as an intermediate complexity snow model. I would classify this as a simple (yet, multilayer) model, whereas Crocus is of intermediate complexity. A complex model is SNOWPACK (Lehning, 2002).

P4, L30: snow pits are two words

P5, L12: SURFEX v8, as in title?

P5, L11: why did you not do bias correction on the radiation data?

P8, L8: 'we increased this value to 600 kg/m3' → only in simulation wind!

P9, L8: models not model

P9, L28: units of thermal insulance are m**2 K / W , see your Figure 7. Units of thermal resistance are K/W. Rename to insulance, or change units to K/W.

P10, formula 5 and 6: parts are missing

P10, L20: this made me wonder, does SURFEX have a representation of snow cover fraction?

P11, L1: Snow height was not well reproduced in 2013-2014 so a 30% reduction to precip has been applied in the model runs. The authors do not discuss the phase of precip. Did you experiment with the temperature threshold for snow?

P11, L2: "in good agreement with the snowpits". How about automatic gauge?

P11, L6: "it seems to be " is not academic English

P11, L12: I'm missing the causal relationship here. Restate like belief or hypothesis.

P11, L30-31: suggest to restate: this partially compensates for the underestimation of density in the upper layers

P12, L5-6: why is the mean density in ES higher than in Crocus? Is this due to the discretization only, or are there differences in the physics?

P12, L9: Suggest to restate: thermal conductivity is primarily controlled by density.

P12, L25: 'gross' is not academic English.

P13, L9: suggest to start the sentence with 'In winter, . . .'

P13, L21-22: suggest to move this to Methods.

P13, L23: unclear

P14, L14: unclear

P14, L17: spelling error 'sate'. Suggest to check entire Latex document using aspell.

P15, L1: 'which confirms' too strong?

P15, L7: warm temperature -> high temperature, 'and at the' → 'at all'

P15, L20: suggest to change 'It is because' to 'This is attributed to'

P15, L27: Rˆ2 is used, yet Table 1 lists R.

P16, L7: suggest to remove 'but', and start sentence with 'Although'

[Figure]

P16, L22-26: discusses temperature, not soil water content.

P16, L29: suggest to drop 'important'

P16, L31: suggest to drop 'considerable'

P17, L1-3: need reference

P17, L5-6: need reference or rewrite.

P17, L8: move TARTES discussion to Methods.

P17, L17: maximum snow density reached by drifting snow

P17, L22: 'main factor influencing soil properties'. Do you mean thermal and/or hydraulic properties?

P17, L28: suggest to change 'increases' to 'enhances'

P18, L19: suggest to use 'ERA-Interim', not 'ERAi'

P19, L2: provide reference to SNOWPACK, Lehning et. Al. (2002)

P19, L12: Crocus is no longer 'most sophisticated'. It is intermediate complexity.

P19, L15: to GCMs

P19, L19: Figures.

P22, L6: Reference Elberling et al. should start on new line.

Figure 7: units for ksnow are misspelled. Suggest singular form in caption (snow height, thermal conductivity). Rename resistance to insulance.

Figure 9: Depth has positive value in the soil. Switch to height if you prefer negative values. No model output between 10 and 20 cm depth, why not include all points that the model has?

---

## Referee Comment (RC2) · Anonymous Referee #2 · 27 Apr 2017

Review of "Evaluating the performance of coupled snow-soil models in SURFEXv8 to simulate the permafrost thermal regime at a high Arctic site" by M. Barrere et al.

The authors present an evaluation of the ability of different combinations of snow and soil developments within the SURFEX model platform, to simulate the snowpack and soil thermal dynamics at a high Arctic site. For that purpose, they collected very detailed and well-thought observations related to both snow and soil. These data are mostly used to validate the snow and soil schemes. Such observations are very rare in Arctic environments, and of great value for model validation. The paper mainly diagnoses forces and weaknesses in the current modelling of snow and soil by schemes of different complexity, one of which is to be run within the CMIP6-experiment. This diagnostic is important for climate modellers in the context of permafrost-carbon climate feedback.

However, first, there are a few shortcomings that undermine the scientific quality of the paper :

* Important literature related to the topic of snow and permafrost is missing. Before the present study, Langer et al. 2013 performed extensive sensitivity tests to assess the critical snow and soil parameters for the soil thermal modelling at a high Arctic permafrost site. Loewe et al. 2013 provided the first 2nd-order bounds for snow thermal conductivity, confirming the importance of anisotropy besides density for the estimation of the Ksnow tensor.

* Together with Referee#1, I noticed that some hypotheses about the origin of biases assessed by the paper, are presented as facts and not hypotheses. Examples of that are : - the early melt erroneously simulated in May 2015 and supposedly caused by lacking zenithal angle dependency in the albedo parametrization (how about missparametrized turbulent fluxes ? local temperature and wind conditions differences with respect to ERA-i ? ...) - the accelerated melt-out induced by albedo parametrization [p15 lines 24-25 : "Ultimately, the treatment of snow albedo in models is such that melt-out is accelerated, so that soil thawing is greatly accelerated in spring"] What is the exact parametrization set responsible for this acceleration, how does it differ from reality ? - the possible impact of vapour flux on the snowpack density could maybe be assessed by simple estimations (order of magnitude)

* Some statements lack the scientific accuracy that should be expected : - a linear regression is used to make ERAi data consistent with original field data; such a regression usually relies on the ordinary least-square method which provides an unbiased estimate. Hence, the statement (p6 line1) that "The correction led to reduce these re-

spective biases by 20%, 3.3% and 10%." is annoying. I believe that the authors meant a reduction in the standard deviation between the corrected ERAi and the observational data. - the explanation of the experimental noise in the ksoil data is vague : is the method really appropriate for frozen soils ? Even though the reader is referred to Domine et al. 2016, a brief assessment of the uncertainty and reliability of the NP method in frozen soils would be welcome here.

A second major point is that the description of the models and in-depth analyses, very much differ between the snow sections (where effects are well described and analysed), and the soil sections (where model description and analysis of phenomena are at times missing). Examples of that are (non exhaustively) listed below : The way organic carbon is included in the soil profile is not clearly described. In brief words, what are the thermal and hydrological consequences of organic carbon in the soils ? What are the thermal properties of the top organic layer featuring litter in ISBA ? Model parameter values (thermal conductivity of the mineral soil, of organic matter, of litter...) should be added to help the reader assess the relevance of the model experiments performed and the added value of the 'increased sophistication'. In the current context, having very fine metrics (Tables 1 and 2) to assess the increase in performance gained by added complexity in the snow and soil schemes, is almost disproportionate. As a result of poor description of the base model and its enhancements with respect to soil processes (litter, SOC), the Results and Discussion sections dedicated to soil are not always informative and omit (or too briefly go through) important potential causes of model errors : i) model parametrization ii) summer soil water content and its impact on the duration of the zero curtain iii) possible impact of the too early snow onset in 2013 on the soil cooling. A thorough improvement of these sections would considerably benefit the paper.

Finally, the manuscript is at times redundant and very precise regarding numeric values, whereas explanations and discussions of the processes could be more developed : A better balance would increase the quality of the paper.

I recommend publication after the mentioned issues have been addressed.

Specific Comments :

* about redundancies : - Whole manuscript : it is maybe not necessary to compare several detailed density or snow values from observations and simulations when the Figures obviously support the fact that the profiles severely differ. This of course belongs the author's choice but it results in an abundance of precision that tend to hide the key message conveyed. - lines 4-12 p 13 exhibit several redundancy and should be reformulated for more synthetic clarity - Lines 6-8 p14 do not add new information w.r.t. preceding text and could be suppressed.

* others : p6 lines 8-10 : which model experiment is compared to snow observations to infer the precipitation correction ?

p9 sect 5.3.1. It should be stated that all experiments except ES are run with the snow model Crocus. I also agree with referee#1 regarding the presentation of these "iterative" experiments. Mind the fact that Fig 4 and 6 have 'Crocus 'instead of 'wind' in their legend (I guess).

p10 line 1 : model-> models. p10 eq 5 and 6 : 'sim' do not appear (probably an edition problem)

p 11 line 14 : the mentioned bottom DH density range [150-200] is not supported by Fig 3 where 2 among 8 densities measured in DH are clearly above 200 kg/m3.

Fig 5: Crocus and ES thermal parametrizations are illustrated here with the same symbol. However, they do not represent the same physical property : while Crocus'ksnow solely accounts for conductive processes, ES-ksnow includes the thermal effects of latent heat fluxes within the snowpack, which makes it an 'apparent' thermal conductivity. My non-expect view on the use of "apparent" vs "effective" for ksnow is the following : "effective" qualifies the 'representative' property of an heterogeneous medium (ref : Gomez-Munoz et al., 2008 ) while "apparent" qualifies the fact that thermal diffusion

[Figure]

through non-conductive processes can sometimes be accounted for with the same law as Fourier diffusion, making it possible to include the relevant diffusion coefficient into the Fourier conductivity. Please correct me on that if I am wrong. Otherwise, I think this key difference in the conductivities of Crocus and ES should be better highlighted on Fig 5 by the use of different symbols and/or in the caption.

References: Gómez-Muñoz, J. L., & Bravo-Castillero, J. (2008). Calculation of effective conductivity of 2D and 3D composite materials with anisotropic constituents and different inclusion shapes in Mathematica. Computer Physics Communications, 179(4), 275-287.

Langer, M., Westermann, S., Heikenfeld, M., Dorn, W., & Boike, J. (2013). Satellite-based modeling of permafrost temperatures in a tundra lowland landscape. Remote Sensing of Environment, 135, 12-24.

Löwe, H., Riche, F., & Schneebeli, M. (2013). A general treatment of snow microstructure exemplified by an improved relation for thermal conductivity. The Cryosphere, 7(5), 1473-1480.

---

## Author Comment (AC1) · 21 Jun 2017

**-Reply to Anonymous Referee #1**

This paper evaluates model skill in reproducing snow and soil characteristics at a permafrost site in Canada. At this site, a variety of in-situ data were retrieved: snow height, snow stratigraphy, temperature, and conductivity. The aim of the study is to find out how good a coupled soil-snow model is at reproducing the transient temperature signal at this site, during two consecutive seasons (2013-2015). Model runs have been performed with the Crocus snow model and the ISBA soil model. Sensitivity experiments are performed one snow property and several soil properties. In addition, the simpler ES snow model has been included, for comparison.

I think this study identifies important shortcomings in these snow models, some of which will apply to other models as well. The article is well structured and contains material that complements existing studies with Crocus. Unfortunately, the paper is not particularly well written and tends to 'jump to conclusions' which undermines its academic quality. I recommend publication after the following issues have been addressed.

*We thank the reviewer for his positive comments and for taking the time to help us optimize our writing. Our responses are embedded in blue italics.*

Major comments

The paper is too long in general. I believe it should fit on 3/4 of the current number of pages. Leave out unnecessary sentences, e.g. only summarize site instrumentation, and refer to full discussion in Domine (2016a). Rewrite long sentences. More use of academic language which is shorter. Focus on main results. Suggest to combine Results and Discussion. Perhaps make two new sections out of those: (1) Snow model results and (2) Soil model results.

*Thank you for the comment. The paper has been shortened following your suggestions. However, we chose to keep the Results and Discussion sections separated to facilitate the readability of the paper and also to avoid confusion as to what is actual result and what is discussion.*

Grammar: the paper could have a better use of linking words to create flow and thus make it easier to read. There are grammatical errors that need to be fixed before publication. I listed some at Specific Comments but I did not aim for completeness.

Suggest to present the experiments in a table, instead of in the text. This would make them easier to refer to and would help to clarify the text, e.g. Sect. 2.3. Then, introduce the experiments earlier in the text, e.g. 'litter' on page 7 line 1, 'SOC' on page 7, line 8-14, 'wind' on page 8, line 8 . What confused me at first is that the additions are additions with respect to other experiments, not 'base'. It would be great if this could be made more clear.

*Thank you for this useful suggestion. We accordingly changed the presentation of the different experiments to make them clearer. Experiments are introduced in the models description, P5, L12: "Based on a series of incremental runs, we particularly focus on the following model features: surface*

*litter, soil organic carbon and density of the drifting snow." And then run 'base' on p.6 l.27, 'litter' on p.7 l.3, 'SOC' on p.7 l.17, 'wind' on p.8 l.8, and 'ES' on p.9 l.11. All the experiments are presented in Table 1. We added details about the iterative experiments (p.9 l.16): "[...] we performed a series of runs with incremental complexities from August 2012 to June 2015 (Table 1). The run wind integrates all our changes, it consists in the most detailed configuration tested here with Crocus. ES uses the same configuration as the run wind."*

One of the main results is Figure 3, that shows that snow density is not well reproduced by the models. It is hypothesized by the authors that this is due to a missing process: upward vapour fluxes. Yet they have no run with this process included so they cannot conclude that this process is fully responsible, only partially, or not at all. The abstract is therefore misleading (Page 1, line 20-22) and should be changed.

*As of today, there is no snow model that describes water vapor transport induced by the temperature gradient. Including this process is a significant research project but is in fact very complex because of the very nature of current snow models that separate the snowpack in several layers of different thicknesses. Efforts are under way but will not lead to a new model for quite a while. That the water vapor flux is important is discussed in detail in Domine et al. (2016a). It is also proven by photographs of shrub branches buried in snow, where depth hoar crystals grow on the underside, demonstrating the existence of a strong flux (Fig. 1). Granted, nothing today proves that this flux is responsible for 100% of the discrepancy between models and observations, but models have other approximations anyway that would make a possible demonstration no more convincing than the observation of intensive crystal growth under shrub branches. In any case, we have modified the abstract (p.1 l.20): "The simulated snow density profiles are unrealistic, which is most likely caused by the lack of representation of the upward water vapour fluxes generated by the strong temperature gradients within the snowpack in snow models."*

[Figure]

Figure 1. Picture of a shrub branch with depth hoar crystals stuck on the underside, demonstrating the existence of a strong upward water vapour flux within the snowpack.

On page 8 line 8 the authors explain that compaction by drifting snow can now reach up to 600 kg/m3, compared to 350 kg/m3 before. The argumentation for this change is anecdotal ('we observed densities of 450 kg/m3'). Should the reader therefore regard this change as just a sensitivity test, rather than a real physical process that was misrepresented? Moreover, doing this you may be compensating for other biases / missing physics in the model, such as the missing upward vapour transport, and, my hypothesis, early melt and refreezing? This potential caveat is not discussed.

*Thank you for these questions. First of all, as stated p17, l. 23, "Numerical models can be viewed as descriptions of a set of complex processes where error compensation is optimized". This is true for Crocus, for ISBA and any other snow or land surface model. And this is why a model used in conditions very different from those it was designed for can lead to large errors: because error compensation is not optimized anymore. At first, as indicated p8, l.6, increasing the maximum density reached by drifting snow concerns a misrepresented physical process, because Arctic snow densities regularly exceed 350 kg m$^{-3}$. In essence, yes, by changing snow density, we are trying to improve error compensation for Arctic conditions, while it is clear that this will in fact attempt to compensate for vertical water transport, which increases upper layer densities. This change was then used as a sensitivity test to assess the impact on snow and soil properties. Changing this parameter may affect other processes, in particular snow driftability and early melt (see next point).*

*This is now stated on p11, l32: "[...] increasing the maximum density reached by drifting snow helps to reduce the underestimation in upper layers." And in the discussion (P17, L1): "This change also partially compensates for vertical water transport, which increases upper layer densities."*

I guess the goal of simulation 'wind' is to simulate a hardened top wind slab. Rather than changing the upper limit to 600 kg/m3 in simulation 'wind', would it not be more effective to decrease the characteristic time scale in the drifting snow compaction (parameter Tau in Vionnet, 2012)? Looking at Figure 3, I see none of the model results exceed 300 kg/m3 at the top, so I wonder if 600 kg/m3 is ever reached at all.

*Changing the characteristic time scale (parameter Tau) was another concern of our work. The results show only slight differences (less than 10%) in snow densities between runs 'wind' (maximum density changed from 350 to 600 kg m$^{-3}$) and 'windTau' (characteristic time scale changed from 48h to 12h), see Fig. 2. Consequences are colder soil temperatures during winter with the run 'windTau', because of the overall more conductive snowpack. Even if the theoretical maximum density of 600 kg m$^{-3}$ is never reached in our simulations, we found interesting to discuss the impact of compensating errors resulting in an apparently well-simulated soil temperature by the run 'wind'. The value 600 kg/m3 is used to change the shape of the relationship between wind speed and snow density, but it does not mean that this value should be reached in the simulations.*

[Figure]

Figure 2. Vertical profiles of density measured on 14 May 2014 (left) and 12 May 2015 (right), and simulated with Crocus on 14 May 2014 and 6 May 2015.

In Figure 3, density in May 2014, the 'wind' experiment simulates lower density at the top than the others. This is counter-intuitive, as you would expect always higher densities in this simulation. Could you provide a possible explanation to why this is?

*Good point. The increase in density of the sub-surface snow layer during spring is due to melt, not to wind-induced compaction. Spring melt is slower in run 'wind' compared to previous experiments, because snow temperatures are lower. This is explained by higher surface thermal conductivities, allowing cold waves to propagate easier to the bottom of the snowpack during winter. In spring, heat waves are also more easily transferred through the snowpack in run 'wind', while in previous experiments heat accumulates in low-conductivity surface snow layers, accelerating their melt-out. A brief explanation is now given on page 11 line 30: "In May 2014, the wind experiment simulates yet lower densities in sub-surface snow layers than the other Crocus runs. This is attributed to early melting, which occurs slower in this run because of colder snow temperatures compared to previous experiments." And in the snow temperature description (P13, L4): "The run wind simulates colder temperatures than previous experiments. This is attributed to the higher conductive upper layers, allowing cold waves to propagate through the snowpack during winter."*

The analysis of snow temperature completely omits the effect of latent heat by rain and meltwater refreezing. What do the authors think is the importance of refreezing on temperature and how do the models simulate this?

*Crocus computes heat fluxes by taking into account latent heat exchanges, including contributions from evaporation of liquid water and sublimation, and melt-refreeze cycle. It also includes a precipitation heat advection term when it is raining. This is detailed in (Vionnet et al., 2012), to which the reader is referred, and it therefore will not be repeated here.*

*Snow melt and water percolation take place in a manner which is spatially very variable. Percolation follows channels in snow, water lenses form very irregularly at discontinuities in capillarities (such as at the base of wind slabs). Simulating this with a 1-D model is a challenge. Observing this using point measurements carries the problem of data representativity. Hence we feel that no interesting discussion on this point can be made here. If we had made an addition, it could only have been "Latent heat exchanges and percolation during snow melt are expected to modify the temperature profile and this is observed in the simulation on 19 May 2015, with the snowpack become almost isothermal. From our measurements, this happens on 10 June. However, given the spatially highly variable nature of percolation, 1-D models and point measurements make any meaningful comparison difficult and of limited interest."*

*It is pretty clear to us that this adds little value to the paper, and for the sake of concision (recommended by the reviewer) we prefer not to develop this point.*

Another key result is Figure 5, that shows that a simple density relationship for thermal conductivity is not sufficient to reproduce most observations. That said, it does not deserve the qualification 'totally inappropriate' (P18, L30).

*Changed to 'inadequate' (P18, L17).*

P17, L5-6: the effect of missing effect of solar zenith angle is stated like a fact. Yet you have no results or reference to support this. Make clear that this is a hypothesis, not a given fact.

*Please see response to referee #2, point 2.*

The two previous comment exemplify a general critique that I have on this article: the wording is not precise enough. In the article, there are sentences without such modifiers that read like facts, but are in reality claims or beliefs of the authors. This must be addressed in the final version.

The authors do not mention whether the model changes they did (litter/SOC/wind) have officially been incorporated into SURFEX.

*Litter and SOC have been integrated to SURFEX v8 (Decharme et al., 2016). Changes we made (maximum snow density value for wind compaction, litter thickness and SOC content) can be easily reproduced.*

Specific comments

P1, L19: soil and snow thermal regime. Simulated soil and snow properties.

*Done.*

P1, L20: compared with → compared to, add comma after 'temperature'

*Done.*

P1, L21: suggest to change 'erroneous' to 'unrealistic'

*Done.*

P1, L29: climate change.

*Done.*

P2, L31: ES is introduced as an intermediate complexity snow model. I would classify this as a simple (yet, multilayer) model, whereas Crocus is of intermediate complexity. A complex model is SNOWPACK (Lehning, 2002).

*Two distinct methods are frequently used to simulate snowpack evolution: the degree-day method used in hydrology and the energy-balance method used in land surface schemes of Numerical Weather Prediction (NWP) and climate models. In the following, we only discuss the complexity of snowpack models using the energy balance method.*

*While there is no official categorization of snowpack models, it is generally agreed that several key components make the delineation between simple, intermediate complexity and sophisticated snowpack models. Simple models are generally single layer schemes used for numerical weather prediction or climate model land surface schemes (see e.g. Essery et al., 1999, Avanzi et al., 2016). Most operational NWP and climate models still use such snow components, while a few of them are gradually moving toward using multi-layer "intermediate complexity" models, with improved representation of heat storage and intrinsic snow processes, which requires multi-layer approaches (see e.g. Essery et al., 2013). ES clearly belongs to this category of models (Decharme et al., 2016).*

*Beyond, there is still a higher class of sophistication for snow models, which in particular includes explicit representations of snow microstructure and sophisticated approaches for handling e.g. solar radiation budget and liquid water percolation. Crocus and SNOWPACK feature approximately the same level of detail for the representation of snow microstructure and its evolution (through the use of semi-empirical variables such as dendricity, sphericity, sometimes replaced by specific surface area - see Carmagnola et al., 2014, and the associated evolution parameterizations), and both handle a large number of snow layers in a Lagrangian way. Liquid water treatment was recently improved in SNOWPACK using the Richards equation (Wever et al., 2014), also more recently implemented in Crocus (D'Amboise et al., 2017). The original version of Crocus uses three spectral bands to compute solar light radiative transfer and albedo, based on physical properties of snow and snow age (Vionnet et al., 2012), while SNOWPACK currently uses a broadband albedo parameterization. Following recent developments, Crocus is now equipped with a spectrally-resolved radiative transfer scheme (Libois et al., 2015), and a multi-physics version of Crocus was recently published (Lafaysse et al., 2017- which includes parameterizations initially implemented in SNOWPACK). Clearly, SNOWPACK and Crocus belong to the same category of snowpack models, with comparably similar levels of sophistication depending on the process.*

*As a consequence, we do not think it is necessary to change the description of ES and Crocus in the manuscript.*

P4, L30: snow pits are two words

*Corrected.*

P5, L12: SURFEX v8, as in title?

*Added.*

P5, L11: why did you not do bias correction on the radiation data?

*As indicated in the paper (p5 l.21), the radiometer shifted from its horizontal position. Therefore, as we added on page 6 line 9: "ERAi radiation data were kept unchanged for lack of reliable measured values"*

P8, L8: 'we increased this value to 600 kg/m3' → only in simulation wind!

*Indicated.*

P9, L8: models not model

*Done.*

P9, L28: units of thermal insulance are m**2 K / W , see your Figure 7. Units of thermal resistance are K/W. Rename to insulance, or change units to K/W.

*Changed to thermal insulance, thanks for this comment.*

P10, formula 5 and 6: parts are missing

*Corrected.*

P10, L20: this made me wonder, does SURFEX have a representation of snow cover fraction?

*SURFEX can indeed represent the surface cover fraction of snow, vegetation and bare ground. Here, we simply used a snow cover fraction of 0 when the surface was snow free, and of 1 when at least 1 mm of snow was covering the ground. We thus do not think it is relevant to add details about it in the paper as what we did on this aspect is pretty clear.*

P11, L1: Snow height was not well reproduced in 2013-2014 so a 30% reduction to precip has been applied in the model runs. The authors do not discuss the phase of precip. Did you experiment with the temperature threshold for snow?

*Changing the temperature threshold of ±1°C for the ERA-interim precipitation phase recalculation lead to changes of 9-10% in cumulated snowfall between August 2013 and August 2015. Because of the high uncertainty on the precipitation rate, and given that observed snow height is quite well reproduced in winter 2014-2015 without changing the snowfall amount or the temperature threshold for precipitation phase, we chose to arbitrarily reduce the snowfall amount for the winter 2013-2014.*

P11, L2: "in good agreement with the snowpits". How about automatic gauge?

*Snow gauge data are shown in Figure 2. As stated in the text, automatic snow gauges give just one point measurement while snow pits yield several data points and are therefore more representative. We feel this was detailed enough in the text, e.g. page 4 line 21, page 10 line 16.*

P11, L6: "it seems to be " is not academic English

*Changed for "which could be".*

P11, L12: I'm missing the causal relationship here. Restate like belief or hypothesis.

*Please see response to referee #2, point 2.*

P11, L30-31: suggest to restate: this partially compensates for the underestimation of density in the upper layers

*Done (P11, L28), thanks for the suggestion.*

P12, L5-6: why is the mean density in ES higher than in Crocus? Is this due to the discretization only, or are there differences in the physics?

*This is due to the faster compaction rate in ES, because it does not account for the snow microstructural property whereas Crocus reduces compaction rates for crystals of depth hoar, frequently found in the Artic snowpack.*

P12, L9: Suggest to restate: thermal conductivity is primarily controlled by density.

*Changed to (P12, L4): "Since thermal conductivity is totally (in Crocus) or mostly (in ES) controlled by density".*

P12, L25: 'gross' is not academic English.

*Changed to "very large".*

P13, L9: suggest to start the sentence with 'In winter, . . .'

*Done.*

P13, L21-22: suggest to move this to Methods.

*We prefer to let it here to facilitate the comprehension.*

P13, L23: unclear

*Changed to (P13, L17): "Litter and SOC additions have little effect on snow properties, the most noticeable being a reduction of less than 1 cm in snow height."*

P14, L14: unclear

*Answered to referee #2, point 5.*

P14, L17: spelling error 'sate'. Suggest to check entire Latex document using aspell.

*Done.*

P15, L1: 'which confirms' too strong?

*Changed to "which supports".*

P15, L7: warm temperature -> high temperature, 'and at the' → 'at all'

*Done.*

P15, L20: suggest to change 'It is because' to 'This is attributed to'

*Changed to "This may be attributed".*

P15, L27: $R^2$ is used, yet Table 1 lists R.

*We are sorry, but Table 1 clearly states r2.*

P16, L7: suggest to remove 'but', and start sentence with 'Although'

*Done.*

P16, L22-26: discusses temperature, not soil water content.

*Removed.*

P16, L29: suggest to drop 'important'

*Removed.*

P16, L31: suggest to drop 'considerable'

*This is what shapes the Arctic snowpack. We do confirm that the impact is considerable.*

P17, L1-3: need reference

*There are already 2 references. The fact that density errors propagate to thermal conductivity is because thermal conductivity is determined solely by density, as already discussed in detail in many places in the paper. We feel there is no need to repeat this here.*

P17, L5-6: need reference or rewrite.

*Rewrote as (P16, L23): "Further, downwelling shortwave absorption is lower in the Arctic because of the large zenith angle in late winter, and this is not accounted for in the original version of Crocus. Therefore, solar warming of the snowpack is exaggerated, resulting in incorrectly simulated melting episodes.*

P17, L8: move TARTES discussion to Methods.

*Done (P8, L21).*

P17, L17: maximum snow density reached by drifting snow

*Done.*

P17, L22: 'main factor influencing soil properties'. Do you mean thermal and/or hydraulic properties?

*Both, changed accordingly (P17, L7).*

P17, L28: suggest to change 'increases' to 'enhances'

*Done.*

P18, L19: suggest to use 'ERA-Interim', not 'ERAi'

*Done.*

P19, L2: provide reference to SNOWPACK, Lehning et. Al. (2002)

*Done.*

P19, L12: Crocus is no longer 'most sophisticated'. It is intermediate complexity.

*See the previous answer.*

P19, L15: to GCMs

*Done.*

P19, L19: Figures.

*Done.*

P22, L6: Reference Elberling et al. should start on new line.

*Done.*

Figure 7: units for ksnow are misspelled. Suggest singular form in caption (snow height, thermal conductivity). Rename resistance to insulance.

*Done.*

Figure 9: Depth has positive value in the soil. Switch to height if you prefer negative values. No model output between 10 and 20 cm depth, why not include all points that the model has?

*Done. No points between 10 and 20 cm depth.*

References:

Avanzi, F., De Michele, C., Morin, S., Carmagnola, C. M., Ghezzi, A., and Lejeune, Y., Model complexity and data requirements in snow hydrology : seeking a balance in practical applications. Hydrol. Process., doi :10.1002/hyp.10782, 2016.

Carmagnola, C. M., Morin, S., Lafaysse, M., Domine, F., Lesaffre, B., Lejeune, Y., Picard, G., and Arnaud, L. : Implementation and evaluation of prognostic representations of the optical diameter of snow in the SURFEX/ISBA-Crocus detailed snowpack model, The Cryosphere, 8, 417-437, doi :10.5194/tc-8-417-2014, 2014.

D'Amboise, C. J. L., Müller, K., Oxarango, L., Morin, S., and Schuler, T. V. : Implementation of a physically based water percolation routine in the Crocus (V7) snowpack model, Geosci. Model Dev. Discuss., doi :10.5194/gmd-2017-56, in review, 2017.

Decharme, B., Brun, E., Boone, A., Delire, C., Le Moigne, P., and Morin, S. : Impacts of snow and organic soils parameterization on northern Eurasian soil temperature profiles simulated by the ISBA land surface model, The Cryosphere, 10, 853-877, doi :10.5194/tc-10-853-2016, 2016.

Domine, F., Barrere, M. and Sarrazin, D.: Seasonal evolution of the effective thermal conductivity of the snow and the soil in high Arctic herb tundra at Bylot Island, Canada, Cryosph., 10(6), 2573–2588, doi:10.5194/tc-10-2573-2016, 2016a.

Lafaysse, M., Cluzet, B., Dumont, M., Lejeune, Y., Vionnet, V., and Morin, S. : A multiphysical ensemble system of numerical snow modelling, The Cryosphere, 11, 1173-1198, doi :10.5194/tc-11-1173-2017, 2017.

Essery R., Martin E., Douville H., Fernandez A., Brun E. (1999), A comparison of four snow models using observations from an Alpine site. Climate Dynamics, Climate Dyn., 15:583-593.

Essery, R., S. Morin, Y. Lejeune, C. B. Ménard. A comparison of 1701 snow models using observations from an alpine site, Adv. Water Resour., 55, 131–148, doi :10.1016/j.advwatres.2012.07.013, 2013.

Libois, Q., Picard, G., Arnaud, L., Dumont, M., Lafaysse, M., Morin, S., and Lefebvre, E. : Summertime evolution of snow specific surface area close to the surface on the Antarctic Plateau, The Cryosphere, 9, 2383-2398, doi :10.5194/tc-9-2383-2015, 2015.

Vionnet, V., Brun, E., Morin, S., Boone, A., Faroux, S., Le Moigne, P., Martin, E., and Willemet, J.-M. : The detailed snowpack scheme Crocus and its implementation in SURFEX v7.2, Geosci. Model Dev., 5, 773-791, doi :10.5194/gmd-5-773-2012, 2012.

Wever, N., Fierz, C., Mitterer, C., Hirashima, H., and Lehning, M.: Solving Richards Equation for snow improves snowpack meltwater runoff estimations in detailed multi-layer snowpack model, The Cryosphere, 8, 257-274, doi:10.5194/tc-8-257-2014, 2014.

---

## Author Comment (AC2) · 21 Jun 2017

**-Reply to Anonymous Referee #2**

Review of "Evaluating the performance of coupled snow-soil models in SURFEXv8 to simulate the permafrost thermal regime at a high Arctic site" by M. Barrere et al.

The authors present an evaluation of the ability of different combinations of snow and soil developments within the SURFEX model platform, to simulate the snowpack and soil thermal dynamics at a high Arctic site. For that purpose, they collected very detailed and well-thought observations related to both snow and soil. These data are mostly used to validate the snow and soil schemes. Such observations are very rare in Arctic environments, and of great value for model validation. The paper mainly diagnoses forces and weaknesses in the current modelling of snow and soil by schemes of different complexity, one of which is to be run within the CMIP6-experiment. This diagnostic is important for climate modellers in the context of permafrost-carbon climate feedback.

*Thank you for these positive comments and for giving us suggestions to improve our paper. Our responses are embedded in blue italics.*

However, first, there are a few shortcomings that undermine the scientific quality of the paper :

* Important literature related to the topic of snow and permafrost is missing. Before the present study, Langer et al. 2013 performed extensive sensitivity tests to assess the critical snow and soil parameters for the soil thermal modelling at a high Arctic permafrost site. Loewe et al. 2013 provided the first 2nd-order bounds for snow thermal conductivity, confirming the importance of anisotropy besides density for the estimation of the Ksnow tensor.

*Thank you for these references. They have been included in our paper (Langer et al., 2013 on p16, l1 and p17, l18; Löwe et al., 2013 on p12, l16 and p18, l18).*

* Together with Referee#1, I noticed that some hypotheses about the origin of biases assessed by the paper, are presented as facts and not hypotheses. Examples of that are : - the early melt erroneously simulated in May 2015 and supposedly caused by lacking zenithal angle dependency in the albedo parametrization (how about missparamatrized turbulent fluxes ? local temperature and wind conditions differences with respect to ERA-i ? ...) - the accelerated melt-out induced by albedo parametrization [p15 lines 24-25 : "Ultimately, the treatment of snow albedo in models is such that melt-out is accelerated, so that soil thawing is greatly accelerated in spring"] What is the exact parametrization set responsible for this acceleration, how does it differ from reality ?

*Further investigations have been conducted using the radiative transfer scheme TARTES implemented in Crocus, resulting in improved simulated melting episodes. This is why we suggest that the early melt is caused by the too simple albedo parameterization, and in particular by the absence of solar zenith angle dependency. We chose not to use TARTES because, as stated on page 8 line 23: "[…]the lack of data on snow impurities (nature, deposition, light-absorbing spectroscopy) prevented us from using TARTES in Bylot Island simulations", as it led to other biases, the discussion of which are beyond the scope of this paper. Also, as stated in our response to Reviewer 1, models rely on error optimizations*

*so that it may be difficult to attribute a given model shortcoming to a single cause. Reasonable suggestion can however be made. In any case, we changed the text as follows:*

*(P11, L8): "We did not observe any signs of spring melt before 18 May during the field campaign. To test whether the inexact melt onset date simulated was caused by the lack of solar zenith angle consideration, we briefly tested Crocus coupled to TARTES, which includes treatment of SZA. With TARTES, the melt onset date was accurately simulated, which leads us to suggest that not accounting for SZA is the main cause of the inadequate melt onset date simulation. However, a full implementation of TARTES in Crocus for this study would have required data on snow impurities."*

*(P15, L14): "Ultimately, the treatment of snow albedo in models is such that melt-out is enhanced under high latitudes, so that soil thawing may be greatly accelerated in spring."*

*(P16, L23): "Further, downwelling shortwave absorption is lower in the Arctic because of the large zenith angle in late winter, and this is not accounted for in the original version of Crocus. Therefore, solar warming of the snowpack is exaggerated, resulting in incorrectly simulated melting episodes."*

-the possible impact of vapour flux on the snowpack density could maybe be assessed by simple estimations (order of magnitude)

*We added on page 16, line 17: "Domine et al. (2016a) estimated vertical water vapor fluxes in the snowpack, and came up with a mass loss of the basal layer of 2.6 kg m$^{-2}$ over two months. It corresponds to a density decrease from 300 to 200 kg m$^{-3}$ for a 3 cm-thick layer, in line with the model overestimation of the basal layer density. Although this estimation is approximate, it supports the suggestion that the vertical water vapor flux is the main cause of the model misrepresentation of the density profile."*

\* Some statements lack the scientific accuracy that should be expected : - a linear regression is used to make ERAi data consistent with original field data; such a regression usually relies on the ordinary least-square method which provides an unbiased estimate. Hence, the statement (p6 line1) that "The correction led to reduce these respective biases by 20%, 3.3% and 10%." is annoying. I believe that the authors meant a reduction in the standard deviation between the corrected ERAi and the observational data.

*Indeed, it was corrected.*

- the explanation of the experimental noise in the ksoil data is vague : is the method really appropriate for frozen soils ? Even though the reader is referred to Domine et al. 2016, a brief assessment of the uncertainty and reliability of the NP method in frozen soils would be welcome here.

*We added on page 5 line 1: "Our instrumental methods impose to use the same heating power in the NPs for snow and soil. The heating power was optimized for snow to minimize heating and hence perturbation to metamorphism. Since soils have a higher thermal conductivity than snow, especially when frozen, the thermal signal of the NPs is low and noise for frozen soil thermal conductivity data is higher than what could be achieved using a higher heating power."*

*And restated (P14, L5): "As detailed above, the noise in the frozen soil data is due to the low power used to heat the NPs."*

A second major point is that the description of the models and in-depth analyses, very much differ between the snow sections (where effects are well described and analysed), and the soil sections (where model description and analysis of phenomena are at times missing). Examples of that are (non exhaustively) listed below : The way organic carbon is included in the soil profile is not clearly described. In brief words, what are the thermal and hydrological consequences of organic carbon in the soils ? What are the thermal properties of the top organic layer featuring litter in ISBA ?

*The parameterization of SOC in ISBA is fully described in Decharme et al., 2016. Soil thermal and hydraulic properties are a combination of organic properties with the standard mineral soil properties. Including SOC affects the vertical profile of soil hydraulic (water retention, matric potential and hydraulic conductivity at saturation) and thermal (thermal conductivity, heat capacity) properties, depending on the organic content in each soil layer (see Fig. 2 in Decharme et al., 2016). We added on page 7, line 9: "Briefly, depending on its content which decreases sharply with soil depth, including organic carbon reduces the dry soil thermal conductivity, increases its porosity and therefore its saturated hydraulic conductivity."*

*About the surface litter, ISBA assigns organic matter thermal properties to uppermost soil layers (P7, L1): "The thermal conductivity of organic matter is set to 0.05 W $m^{-1}$ $K^{-1}$ when dry, and to 0.25 W $m^{-1}$ $K^{-1}$ when wet."*

*Figure 9 shows differences in soil thermal conductivity and water content profiles before and after organic carbon additions (run litter, SOC and following).*

Model parameter values (thermal conductivity of the mineral soil, of organic matter, of litter...) should be added to help the reader assess the relevance of the model experiments performed and the added value of the 'increased sophistication'.

*Details and values were added on page 6 line 28: "The soil thermal properties, i.e. thermal conductivity and heat capacity, are computed as a combination of water, ice and soil properties, volumetric water content and soil porosity, following the parameterizations of Peters-Lidard et al. (1998). Hence, the soil thermal conductivity is expressed as a function of its saturation, porosity, quartz content, dry soil conductivity and phase of water (frozen or unfrozen), where the ice, water and quartz thermal conductivities are respectively 2.2, 0.57 and 7.7 W $m^{-1}$ $K^{-1}$."*

In the current context, having very fine metrics (Tables 1 and 2) to assess the increase in performance gained by added complexity in the snow and soil schemes, is almost disproportionate.

*Indeed, the accuracy shown in Tables 2 and 3 ($10^{-2}$) could be judged as excessive regarding the current context. However, it helps to distinguish differences between experiments, to identify important processes involved when simulating the permafrost thermal regime.*

As a result of poor description of the base model and its enhancements with respect to soil processes (litter, SOC), the Results and Discussion sections dedicated to soil are not always informative and omit (or too briefly go through) important potential causes of model errors : i) model parametrization ii) summer soil water content and its impact on the duration of the zero curtain iii) possible impact of the too early snow onset in 2013 on the soil cooling. A thorough improvement of these sections would considerably benefit the paper.

*We tried to improve the description of soil model parameterizations, as you suggested. Concerning the impact of summer soil water content on the duration of the zero curtain, we added details on page 15 line 24: "The main difference with observation is the timing of freezing and thawing. This is clearly influenced by temperature, so that the errors in simulated temperature impact this timing. However, the duration of latent heat exchanges are also determined by the water content." Then on page 15 line 32: "[...] it seems that the simulation of the water dynamics in the first cm of the soil is erroneous, and is better reproduced below -10 cm. The low simulated VWC values therefore partly explain the too fast simulated soil freezing. However, as discussed by Langer et al., (2013), the inadequate thermal properties of the snow cover is most likely the main source of error in the ground thermal regime."*

*About the early snow onset in 2013, it accelerates soil cooling because of the simulated high basal $k_{snow}$ values (P15, L9): "This may be attributed to the thermal conductivity of the basal snow layer which is too high in simulations, allowing the soil to cool rapidly."*

Finally, the manuscript is at times redundant and very precise regarding numeric values, whereas explanations and discussions of the processes could be more developed: A better balance would increase the quality of the paper. I recommend publication after the mentioned issues have been addressed.

Specific Comments :

* about redundancies : - Whole manuscript : it is maybe not necessary to compare several detailed density or snow values from observations and simulations when the Figures obviously support the fact that the profiles severely differ. This of course belongs the author's choice but it results in an abundance of precision that tend to hide the key message conveyed. - lines 4-12 p 13 exhibit several redundancy and should be reformulated for more synthetic clarity

*Done.*

- Lines 6-8 p14 do not add new information w.r.t. preceding text and could be suppressed.

*Done.*

* others : p6 lines 8-10 : which model experiment is compared to snow observations to infer the precipitation correction ?

*Any Crocus experiment can be used to infer the precipitation correction, as they are very slight differences in snow height (see Fig. 7). Added on p6 l.6: "we arbitrarily changed the ERAi precipitation data for Crocus to match the observed snow height at snow pits dug in the immediate vicinity of BylSta"*

p9 sect 5.3.1. It should be stated that all experiments except ES are run with the snow model Crocus. I also agree with referee#1 regarding the presentation of these "iterative" experiments. Mind the fact that Fig 4 and 6 have 'Crocus 'instead of 'wind' in their legend (I guess).

*Done. Fig. 4 and 6 corrected.*

p10 line 1 : model-> models. p10 eq 5 and 6 : 'sim' do not appear (probably an edition problem)

*Corrected.*

p 11 line 14 : the mentioned bottom DH density range [150-200] is not supported by Fig 3 where 2 among 8 densities measured in DH are clearly above 200 kg/m3.

*The given density range [150-200 kg m$^{-3}$] reflects the very bottom snow layer (or basal layer) only, the layer at the interface with the ground. On the 2 profiles shown it is about 5 cm-thick. The 2 densities measured above 200 kg m$^{-3}$ were taken in the depth hoar layer situated above the basal indurated depth hoar layer.*

Fig 5: Crocus and ES thermal parametrizations are illustrated here with the same symbol. However, they do not represent the same physical property : while Crocus' ksnow solely accounts for conductive processes, ES-ksnow includes the thermal effects of latent heat fluxes within the snowpack, which makes it an 'apparent' thermal conductivity. My non-expect view on the use of "apparent" vs "effective" for ksnow is the following: "effective" qualifies the 'representative' property of an heterogeneous medium (ref : Gomez-Munoz et al., 2008 ) while "apparent" qualifies the fact that thermal diffusion through non-conductive processes can sometimes be accounted for with the same law as Fourier diffusion, making it possible to include the relevant diffusion coefficient into the Fourier conductivity. Please correct me on that if I am wrong. Otherwise, I think this key difference in the conductivities of Crocus and ES should be better highlighted on Fig 5 by the use of different symbols and/or in the caption.

*Yes indeed, ES calculates an apparent thermal conductivity. We followed your suggestion and changed the symbol for ES. We also added the following sentence in the caption: "Crocus computes k$_{snow}$ from the density only following Yen's parameterization, while ES includes the thermal effects of latent heat fluxes within the snowpack."*

References:

Decharme, B., Brun, E., Boone, A., Delire, C., Le Moigne, P. and Morin, S.: Impacts of snow and organic soils parameterization on northern Eurasian soil temperature profiles simulated by the ISBA land surface model, Cryosph., 10(2), 853–877, doi:10.5194/tc-10-853-2016, 2016.

Domine, F., Barrere, M. and Sarrazin, D.: Seasonal evolution of the effective thermal conductivity of the snow and the soil in high Arctic herb tundra at Bylot Island, Canada, Cryosph., 10(6), 2573–2588, doi:10.5194/tc-10-2573-2016, 2016a.

Gómez-Muñoz, J. L., & Bravo-Castillero, J. (2008). Calculation of effective conductivity of 2D and 3D composite materials with anisotropic constituents and different inclusion shapes in Mathematica. Computer Physics Communications, 179(4), 275-287.

Langer, M., Westermann, S., Heikenfeld, M., Dorn, W., & Boike, J. (2013). Satellite-based modeling of permafrost temperatures in a tundra lowland landscape. Remote Sensing of Environment, 135, 12-24.

Löwe, H., Riche, F., & Schneebeli, M. (2013). A general treatment of snow microstructure exemplified by an improved relation for thermal conductivity. The Cryosphere, 7(5), 1473-1480.

Peters-Lidard, C. D., Blackburn, E., Liang, X. and Wood, E. F.: The effect of soil thermal conductivity parameterization on surface energy fluxes and temperatures, J. Atmos. Sci., 55, 1209– 1224, 1998.

---

## Referee Report (RR1)

I thank the authors for their corrections and substantial manuscript improvement.

I am afraid I missed a likely erroneous statement in my original review, which the authors just rephrased in the corrected manuscript : "To a first approximation, under steady state the temperature gradient is inversely proportional to ksnow. Since the simulated thermal conductivity profile is inverted, we expect simulated snow surface temperatures to be colder than measurements while bottom temperatures should be warmer, given that simulated and measured temperatures are similar near the middle of the snowpack" (p28 in the authors'response).

An inverted Keff profile in simulations should indeed lead to colder surface temperatures but also colder soil temperatures (at the bottom, Keff is higher-than-in-real, so temperature gradient is weak and departure from the 22cm T°C value is low, leading to colder-than-real bottom snow temperature). What is seen fig 6, -- ie that crocus (eg : wind) simulations are colder than observations at the surface, and warmer at the bottom of the snowpack -- is probably more an effect of different arithmetic Keff averages between observations and simulations.

I think the manuscript is now ready for publications, pending that the authors address this minor and last comment.

---

## Author Response (AR2)

I thank the authors for their corrections and substantial manuscript improvement.

I am afraid I missed a likely erroneous statement in my original review, which the authors just rephrased in the corrected manuscript : "To a first approximation, under steady state the temperature gradient is inversely proportional to ksnow. Since the simulated thermal conductivity profile is inverted, we expect simulated snow surface temperatures to be colder than measurements while bottom temperatures should be warmer, given that simulated and measured temperatures are similar near the middle of the snowpack" (p28 in the authors'response).

An inverted Keff profile in simulations should indeed lead to colder surface temperatures but also colder soil temperatures (at the bottom, Keff is higher-than-in-real, so temperature gradient is weak and departure from the 22cm T°C value is low, leading to colder-than-real bottom snow temperature). What is seen fig 6, -- ie that crocus (eg : wind) simulations are colder than observations at the surface, and warmer at the bottom of the snowpack -- is probably more an effect of different arithmetic Keff averages between observations and simulations.

I think the manuscript is now ready for publications, pending that the authors address this minor and last comment.

*Thank you for taking the time to correct the paper. Our statement was indeed erroneous, an inverted ksnow profile should result in a colder snow-soil interface temperature. We rewrote this section as following (P12, L32):*

[revised manuscript text omitted]